# Antibodies to the RBD of SARS-CoV-2 spike mediate productive infection of primary human macrophages

Suzanne Pickering [1] ✉, Harry Wilson [1], Enrico Bravo [1], Marianne R. Perera[1], Jeffrey Seow [1], Carl Graham[1], Nathalia Almeida[1], Lazaros Fotopoulos[2,3], Thomas Williams[2,3], Atlanta Moitra [1], Helena Winstone[1], Tinne A. D. Nissen [1,4], Rui Pedro Galão [1], Luke B. Snell [1,5], Katie J. Doores [1], Michael H. Malim [1] & Stuart J. D. Neil [1]

The role of myeloid cells in the pathogenesis of SARS-CoV-2 is well established, in particular as drivers of cytokine production and systemic inflammation characteristic of severe COVID-19. However, the potential for myeloid cells to act as bona fide targets of productive SARS-CoV-2 infection, and the specifics of entry, remain unclear. Using a panel of anti-SARS-CoV-2 monoclonal antibodies (mAbs) we performed a detailed assessment of antibody-mediated infection of monocytes/macrophages. mAbs with the most consistent potential to mediate infection were those targeting a conserved region of the receptor binding domain (RBD; group 1/class 4). Infection was closely related to the neutralising concentration of the mAbs, with peak infection occurring below the IC50, while pre-treating cells with remdesivir or FcγRI-blocking antibodies inhibited infection. Studies performed in primary macrophages demonstrated high-level and productive infection, with infected macrophages appearing multinucleated and syncytial. Infection was not seen in the absence of antibody with the same quantity of virus. Addition of ruxolitinib significantly increased infection, indicating restraint of infection through innate immune mechanisms rather than entry. High-level production of pro-inflammatory cytokines directly correlated with macrophage infection levels. We hypothesise that infection via antibody-FcR interactions could contribute to pathogenesis in primary infection, systemic virus spread or persistent infection.

Severe COVID-19 is characterised by immune dysregulation, acute respiratory distress syndrome (ARDS) and multiorgan dysfunction, with the risk of severe disease increasing with age and a range of comorbidities[1]. Biomarkers for multisystem inflammation include elevated tissue damage markers, inflammatory markers, proinflammatory cytokines including IL-6, IL-1, CXCL10 and IL-10, and lymphopenia[1–5]; while viral RNA detected in the blood remains one of the most accurate early indicators of COVID-19 mortality[6–10]. Treatment options during the early waves of the pandemic relied on administering supplementary oxygen and suppressing inappropriate

[1]Department of Infectious Diseases, School of Immunology & Microbial Sciences, King's College London, London SE1 9RT, UK. [2]The Stem Cell Hotel, King's College London, Guy's Hospital, Floor 28, Tower Wing, Great Maze Pond, London SE1 9RT, UK. [3]Centre for Gene Therapy and Regenerative Medicine, King's College London, Guy's Hospital, Floor 28, Tower Wing, Great Maze Pond, London SE1 9RT, UK. [4]Department of Basic and Clinical Neuroscience, Institute of Psychiatry, Psychology and Neuroscience, King's College London, London SE5 9RT, UK. [5]Centre for Clinical Infection and Diagnostics Research, Department of Infectious Diseases, Guy's and St Thomas' NHS Foundation Trust, London SE1 7EH, UK. ✉e-mail: Suzanne.pickering@kcl.ac.uk

and detrimental inflammatory responses using the corticosteroid dexamethasone[1].

Myeloid cells are known to play a central role in the perpetuation of excessive inflammatory responses seen in severe disease and are substantially dysregulated throughout the COVID-19 disease course[2,11–16]. They produce high levels of proinflammatory cytokines and have been associated with inflammasome activation in severe disease[17]. In COVID-19 postmortem studies, bronchoalveolar lavage studies and animal models, monocytes/macrophages are over-represented in the lung[11,14,18–20] and associations have been found between CCR2 expression in the lung, which promotes chemotaxis of monocytes/macrophages to the site of infection via the ligand MCP-1, and disease severity[20,21]. They are frequently found to contain viral proteins and RNA[11,12,14,19,22–26] and exhibit aberrant activation profiles[12,13,19,20]. Furthermore, atypical monocyte activity is implicated in the rare inflammatory syndrome in children, MIS-C, which is associated with SARS-CoV-2 infection[27].

Thus, the role of myeloid cells in the immunopathology of COVID-19 is well established. However, whether there is a requirement for them to become infected by SARS-CoV-2 in order to exert these effects, and importantly, whether the infection is productive or abortive is less clear. In a study of fresh PBMCs isolated from COVID-19 patients, Junqueira et al. found that approximately 6% of peripheral blood monocytes showed evidence of SARS-CoV-2 infection, and that monocytes from COVID-19 patients had increased levels of inflammasome activation, with pyroptosis biomarkers in plasma (GSDMD, LDH, IL-1RA and IL-18) increasing with COVID-19 disease severity[17]. Experimentally, it is difficult to distinguish between an actively infected myeloid cell and one that has engulfed an infected cell or viral/cellular debris, particularly in the case of flow cytometric and RNA-seq approaches. Human monocytes and macrophages do not express ACE2, the primary receptor for SARS-CoV-2[17,19,22,28], and are refractory to infection in vitro[29,30], thus are not classical cellular targets for infection. Yet, an increasing number of reports suggest that myeloid cells are directly infected in vivo, and that this infection is mediated not by typical SARS-CoV-2 receptors but by interactions between Fc receptors on the target cell and antibodies bound to the virus[17,22,25,31,32], as has been previously established for SARS, MERS and other coronaviruses[33–39].

Here we investigate the potential for antibodies to mediate productive SARS-CoV-2 infection of primary human macrophages. We present a detailed assessment of the type of antibodies required to mediate infection, the relationship between the promotion of infection and neutralisation, the productivity of infection in monocytes and primary macrophages and the resolution of this infection by innate immune mechanisms. Furthermore, we demonstrate that individual infection-promoting antibodies are rendered impotent against a backdrop of a fully developed robust polyclonal antibody response.

## Results

### Monoclonal antibodies with defined binding specificities to the SARS-CoV-2 spike RBD can mediate infection of monocytes

To document the relationship between neutralisation and potentiation of infection, we established a model system based on SARS-CoV-1 and mAb CR3022[40], in which parallel assays were performed on permissive HeLa-ACE2 epithelial cells and non-permissive THP-1 monocytic cells (Supplementary Fig. 1a). A panel of 28 mAbs with previously determined specificities for SARS-CoV-2 spike[41–43], were assessed for their neutralising and infection-promoting properties using HIV-1-based pseudoviruses bearing SARS-CoV-2 wildtype D614G spikes (B.1 lineage; Fig. 1a and b). All mAbs were IgG1, and were representative examples from 7 distinct binding groups: groups 1-4 binding to the RBD, 5 and 6 to the NTD, and 7 to SD1 (Supplementary Fig. 1b)[41–43]. The majority of the RBD-specific mAbs demonstrated a strictly concentration-dependent ability to promote infection of THP-1 cells, with the

magnitude of infection varying widely between the mAbs, peaking at almost 2000-fold above background (for group 1 mAb VA14_R39, Fig. 1a and b). In the case of neutralising mAbs, peak infection levels typically occurred at or just below the IC50 and declined with increasing mAb concentration. Comparison of peak infection levels mediated by each mAb, arranged by binding group, demonstrates the consistency of the group 1 and 2 mAbs in promoting infection of THP-1 cells (Fig. 1b), whereas group 3 mAbs were able to mediate infection but at a level significantly lower than group 1. Group 4 mAbs showed a mixed profile, with some mediating very high level THP-1 infection and some no signal at all (Fig. 1b). Anti-NTD and -SD1 mAbs did not mediate infection. No infection was seen in THP-1 cells in the absence of antibody, despite the use of equivalent input of pseudovirus. This observation is reinforced by negligible levels of infection in the presence of increasing levels of concentrated pseudovirus (Supplementary Fig. 1c); in contrast, THP-1 infection in the presence of mAb approach levels seen in the permissive HeLa-ACE2 cells. Some mAbs were extremely effective at mediating infection, with luciferase levels comparable to, and even exceeding, those seen in the permissive HeLa-ACE2 cells (Supplementary Fig. 1d), indicating that this represents an efficient entry pathway for SARS-CoV-2 when experimental conditions are optimal.

Performing HeLa-ACE2 and THP-1 assays in parallel allowed a precise comparison of the neutralising and infection-mediating profiles of the individual mAbs. The mAb concentration at which peak infection occurred in the THP-1 cells was tightly correlated to the IC50 of the individual mAb (Fig. 1c), whereas the potency of neutralisation was not linked to the ability of the mAb to promote infection (Fig. 1d).

Previous studies have shown that antibody-mediated infection of FcγR-expressing cells by SARS-CoV-2 can occur via the major IgG receptors CD64 (FcγRI)[44], CD32 (FcγRII)[31,44] and CD16 (FcγRIII)[17]. To confirm that infection of THP-1 cells is mediated by FcRs, and to narrow down the subtype, we performed Fc receptor blocking experiments by pre-incubating cells with individual FcγR function-blocking mAbs prior to antibody-mediated infection. While THP-1 cells do express a robust level of CD32 and CD64 (Supplementary Fig. 1e), they do not express CD16 (FcγRIII), therefore this mAb served as an additional control. Blocking CD64 significantly reduced infection levels, by over 94% (Fig. 1e). Blocking CD32 had a reproducible but non-significant effect on infection. None of the FcR-blocking mAbs had a significant effect on identical infection conditions in HeLa-ACE2 cells.

We next extended our findings to include SARS-CoV-2 spikes from wildtype (B.1) virus without the D614G mutation and early wave variants alpha (B.1.1.7) and delta (B.1.617.2), plus early omicron variant BA.1 (B.1.1.529.1). Using representative mAbs from RBD binding groups 1-4, parallel HeLa-ACE2 and THP-1 assays were performed (Fig. 2). The group 1 and 2 mAbs promoted infection of wildtype, D614G, delta and to a lesser extent alpha, but had minimal effect on omicron. The latter effect may be due to reduced binding as a consequence of a triple serine mutation occurring in the group 1 binding site at the base of the RBD. The group 3 mAb, while still neutralising and thus able to bind all variants, demonstrated lower potency of infection, typical for mAbs from this group. The group 4 mAb P008_015 was able to promote high-level infection of all variants tested. Interestingly, and in agreement with results shown in Fig. 1a and c, reduced neutralisation potency of P008_015 against delta and omicron was accompanied by a concomitant shift in the concentration at which peak THP-1 infection occurred. A second group 4 mAb, P008_087, neutralised all variants but was unable to mediate infection of THP-1 cells.

To expand the finding that escape from neutralising antibody leads to a shift in the quantity of mAb required to achieve peak infection of THP-1 cells, we used PMS20, a synthetic spike designed to have maximal wave 1 polyclonal neutralisation escape[45]. Unlike

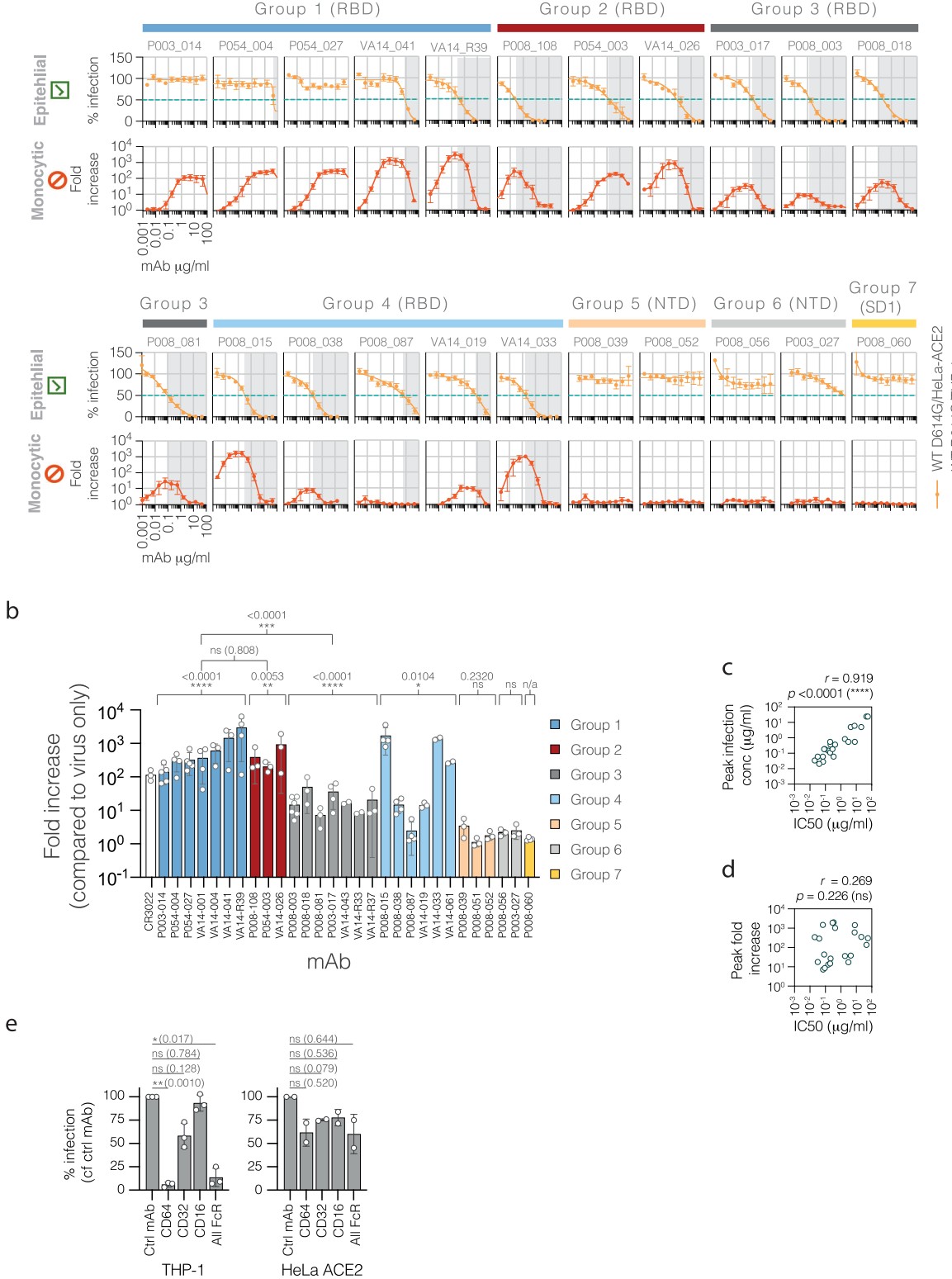

omicron, PMS20 does not have mutations in the group 1 binding region of RBD[45–47]. Thus, PMS20 and the wildtype R683G spike[45] were not neutralised by P054_027 and P003_014, but were both able to infect THP-1 cells in the presence of these mAbs, similar to levels seen for wildtype, wildtype D614G and delta pseudoviruses (Supplementary Fig. 1f and Fig. 2). Escape from neutralisation by mAbs P008_108 and P008_015 by PMS20 corresponded with a shift in peak infection from

0.185 µg/ml for wildtype to 5 µg/ml for PMS20 for mAb P008_108 and 0.185 µg/ml for wildtype to 15 µg/ml for PMS20 for mAb P008_015.

Together these results demonstrate that a subset of non-neutralising anti-RBD mAbs, or neutralising mAbs at sub-neutralising concentrations, can expand the tropism of SARS-CoV-2 to monocytic cells, frequently at infection levels comparable to those seen in typical ACE2-mediated infection of permissive cells.

**Fig. 1 | Monoclonal antibodies with defined binding specificities to the SARS-CoV-2 spike RBD can mediate infection of monocytes. a** Assays were performed in parallel on SARS-CoV-2-permissive epithelial cells (HeLa-ACE2, indicated by green tick) and non-permissive monocytic cells (THP-1, indicated by red no entry symbol). Pseudoviruses with wildtype D614G SARS-CoV-2 spikes were pre-incubated with mAbs before addition to cells, with RLU measured 48 h later. % infection and fold increase were calculated relative to infection levels in the absence of antibody. Grey shaded areas indicate mAb concentrations at which greater than 50% neutralisation occured. Representative examples from seven spike binding specificity groups are shown (RBD groups 1–4, NTD groups 5 and 6, SD1 group 7). Means are from three independent experiments, error bars ±SEM. **b** Summary of infection-promoting potential of 28 mAbs, displayed as peak infection (fold increase relative to no mAb) in THP-1 assays. Bars are coloured according to spike binding specificity. Peak infection level of SARS-CoV-1 mediated by mAb CR3022 is shown for comparison (white bar). Means are from at least two independent experiments, error bars ±SD. Results for each binding specificity group

were compared to theoretical mean of 1 (no infection) using two-tailed one sample t-tests. Results from group 1 were compared with other groups using two-tailed Welch's t-tests: >0.1 (ns), <0.1 (*), <0.01 (**), <0.001 (***), <0.0001 (****). **c** The mAb concentration at which peak infection of THP-1 monocytes occurred and (**d**) the peak magnitude of THP-1 infection (measured by fold increase in the presence vs absence of antibody) were compared with IC50 values for all neutralising RBD-specific mAbs that mediated THP-1 infection ($n = 22$). $r$ and $p$ values were determined by two-tailed Spearman correlation, >0.1 (ns), <0.1 (*), <0.01 (**), <0.001 (***), <0.0001 (****). **e** FcγR blocking experiments were performed by pre-incubating THP-1 cells with function-blocking mAbs for FcγRI (CD64), FcγRII (CD32) or FcγRIII (CD16), or isotype control mAb, then measuring infection mediated by mAb P003_014. Parallel HeLa-ACE2 assays were performed as controls. Means are from three (THP-1) or two (HeLa ACE2 control cells) independent experiments, error bars ±SD. Infection levels in the presence of FcR blocking mAb were compared to control mAb for each condition using two-way ANOVA with mixed effects analyses: >0.1 (ns), <0.1 (*), <0.01 (**), <0.001 (***), <0.0001 (****).

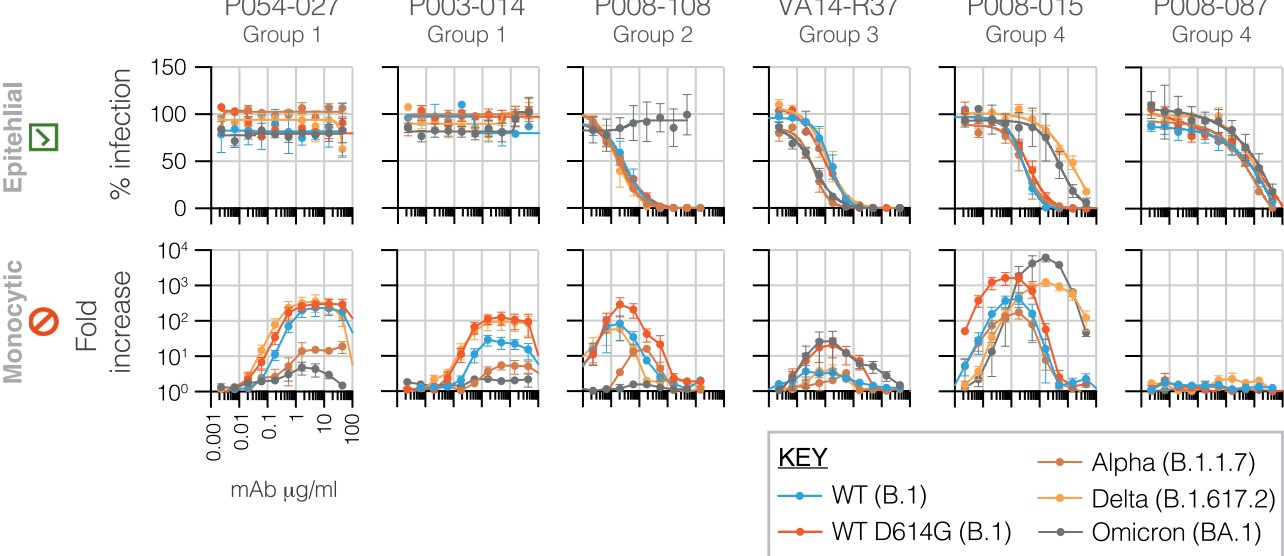

**Fig. 2 | SARS-CoV-2 RBD-specific mAbs promote infection of several VOCs.** Representative mAbs from each of the four RBD binding groups shown in A were tested in parallel HeLa-ACE2 and THP-1 assays, as for A, using pseudoviruses with spikes from wildtype (WT) SARS-CoV-2 (B.1), wildtype with the D614G mutation (WT D614G; B.1) and variants of concern alpha (B.1.1.7), delta (B.1.617.2) and omicron (B.1.1.529). Means are derived from three independent experiments, error bars ±SEM.

## SARS-CoV-2 RBD-specific mAbs promote productive infection of monocytic cells

For both SARS-CoV-1 and −2, infection of myeloid cells has been characterised as abortive, with virus entering the cell but unable to complete a replicative cycle[17,25,33,34,48,49]. Thus, to address the question of whether antibody-mediated infection of monocytic cells is productive, experiments were performed with infectious SARS-CoV-2. Wildtype SARS-CoV-2 (lineage B.1; strain England 02) and variants of concern (VOCs) alpha (B.1.1.7), beta (B.1.351), delta (B.1.617.2) and omicron (BA.1) were incubated with two group 1 infection-promoting mAbs (P003_014 and P054_027) and one group 4 non- infection-promoting mAb (P008_087) prior to addition to THP-1 cells. Cells were intracellularly stained for nucleocapsid 48 hours after infection. Similar to results shown for pseudoviruses, no infection was seen in the absence of mAb (Fig. 3a), with the percentage of nucleocapsid-positive cells for these conditions similar to controls containing no virus. Likewise, no infection was seen in the presence of the control mAb P008_087. However, significant infection levels were observed for wildtype, alpha, beta and delta viruses in the presence of mAbs P014_003 and P054_027 (Fig. 3a), with delta showing the highest levels

of infection (averaging 28.6% in the presence of mAb P054_027, with individual replicates reaching 56.7%).

Active viral replication, rather than viral uptake through FcR-mediated phagocytosis, was strongly indicated both by the absence of signal detected in the control mAb P008_087 cultures and by the 2-log shift in intracellular nucleocapsid detection in infected cells, shown in representative flow cytometry plots in Fig. 3a. This was confirmed by the measurement of infectious virus in the THP-1 cell culture supernatant by plaque assay. The quantity of infectious virus detected was proportional to the percentage of infected cells (Fig. 3b). In the absence of mAb, residual input virus detected at 72 hours was equivalent for all 5 variants, whereas no virus was detected in the cultures containing mAb P008_087. Pre-treatment of THP-1 cells with remdesivir, an inhibitor of SARS-CoV-2 replication, completely abolished infection (Fig. 3c), thus excluding the possibility that phagocytic virus uptake accounted for the nucleocapsid staining observed. Additional experiments performed in PMA-differentiated THP-1 cells, as a proxy for monocyte-derived macrophages, demonstrated results consistent with undifferentiated THP-1 cells (Supplementary Fig. 2a). Intracellular staining for SARS-CoV-2

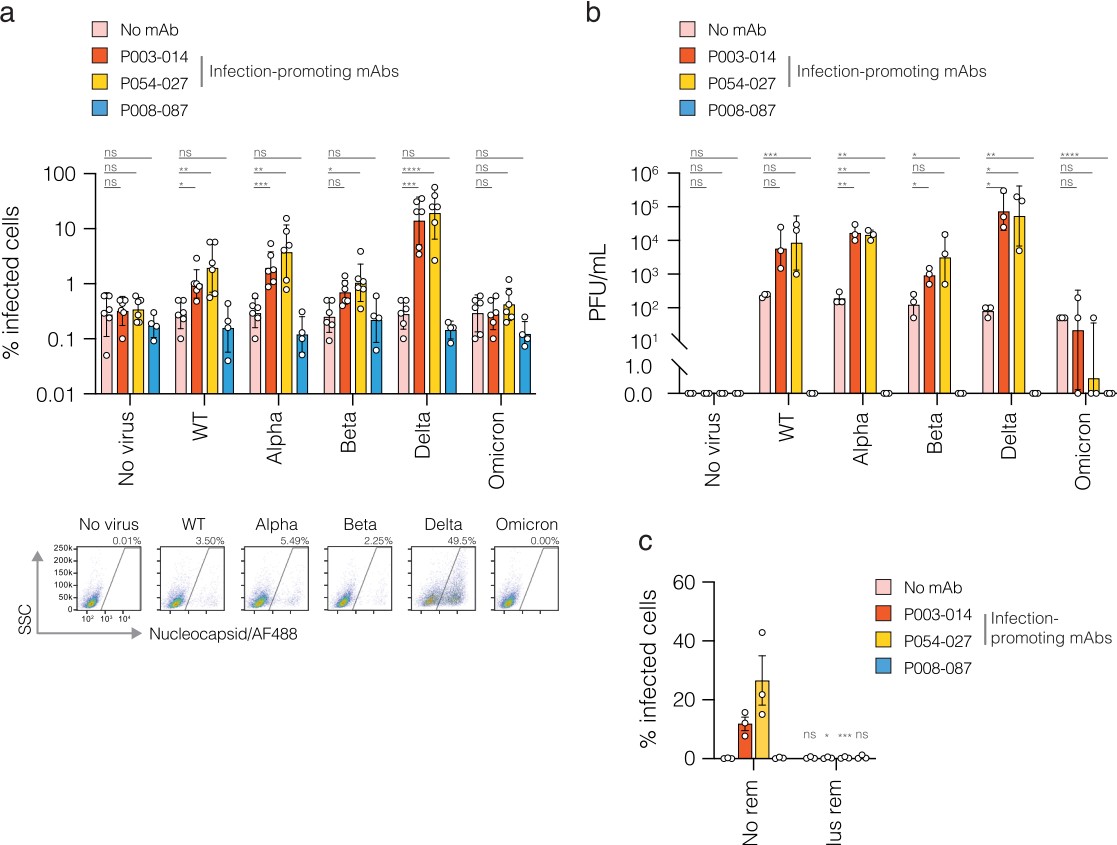

**Fig. 3 | SARS-CoV-2 RBD-specific mAbs promote productive infection of monocytes.** Five major variants of infectious SARS-CoV-2 – wildtype (WT; B.1), alpha (B.1.1.7), beta (B.1.351), delta (B.1.617.2) and omicron (B.1.1.529.1) – were compared for the ability to infect monocytes. Viruses were preincubated with infection-promoting mAbs P003_014, P054_027 or non-infection-promoting control mAb P008_087 at 6 µg/ml prior to addition to THP-1 cells at an MOI of 1. 72 hours later cells were intracellularly stained for SARS-CoV-2 nucleocapsid and % infected cells measured by flow cytometry. Control cultures containing no virus or no mAb were included in all experiments. Dotted line shows the highest mean value obtained in the absence of virus (0.28%), thus representing background cut-off. Means were derived from 6 independent experiments for no mAb, P003_014 and P054_027 and 4 independent experiments for the control mAb P008_087, with error bars representing ±SD. Two-way ANOVA with Tukey's multiple comparison was used to determine whether infection was significant in the presence vs absence of mAb for each virus. >0.1 (ns), <0.1 (*), <0.01 (**), <0.001 (***), <0.0001 (****). Representative flow cytometry plots showing % nucleocapsid positive cells in the presence of P054_027 mAb are shown below each variant and no virus control. **b** Cell culture supernatants from the THP-1 infections shown in (**a**) were assayed for the presence of infectious virus by Vero.E6.TMPRSS2 plaque assay. Mean PFU/mL were calculated from three independent experiments, error bars ±SD. To enable values of zero to be shown on the same graph, the bottom portion of the y-axis has been converted to linear. Two-way ANOVA was used to compare virus quantities measured in the cell culture supernatant in the presence of each mAb vs no mAb for each variant. >0.1 (ns), <0.1 (*), <0.01 (**), <0.001 (***), <0.0001 (****). **c** THP-1 cells were pre-treated with remdesivir (10 µM) then infected as in (**a**) with delta virus at an MOI of 1, pre-incubated with mAbs P003_014, P054_027 or P008_087 or no mAb. 72 hours later cells were intracellularly stained for SARS-CoV-2 nucleocapsid and % infected cells measured by flow cytometry. Means from three independent experiments, error bars ±SEM. Two-way ANOVA was used to compare infection levels with and without remdesivir for each condition. >0.1 (ns), <0.1 (*), <0.01 (**), <0.001 (***), <0.0001 (****).

nucleocapsid demonstrated cytoplasmic distribution and cells appeared syncytial and multinucleate, indicative of ongoing virus replication (Supplementary Fig. 2a).

To further substantiate the notion that monocyte infection is productive, time course experiments were performed in which THP-1 cells were infected with mAb-opsonised wildtype, alpha or delta SARS-CoV-2 and harvested at regular intervals up to 7 days (Fig. 4). In the absence of mAb, presence of control mAb P008_087 or absence of virus, the percent nucleocapsid positive cells remained consistently at background levels for the duration of the experiment. In the presence of infection-promoting mAbs P003_014 and P054_027, however, percent positive cells rose to peak at day 3 for wildtype and alpha, and day 2 for delta. For the latter, the percent infected cells reached a mean of 36.9% (individual replicates reached 52.1%).

Infectious virus in the supernatant was equivalent for all conditions at day 0, but steadily declined over the 7-day time course in the

absence of mAb, consistent with natural decay of the input virus over time and similar for all three viral variants (Fig. 4). In the presence of control mAb P008_087, this decay was accelerated, potentially due to active clearance of the input virus due to antibody-mediated phagocytosis by the THP-1 cells. In the presence of mAb P054_027, an eclipse phase can be detected wherein virus levels initially decline, then viral production is boosted above the level of initial input virus, peaking at day 3 for wildtype and alpha and day 2 for delta. Results for P003_014, which promoted pseudoviral infection to a lesser extent than P054_027 for all three variants (Fig. 2), mirrored this pattern for alpha and delta but to an overall lower level.

The increased detection of infectious virus above the level of input virus in the presence of infection-promoting mAbs—while in parallel the detection of virus in the presence of control mAb P008_087 rapidly declined to undetectable levels for all variants—demonstrates productive infection of monocytic cells by SARS-CoV-2.

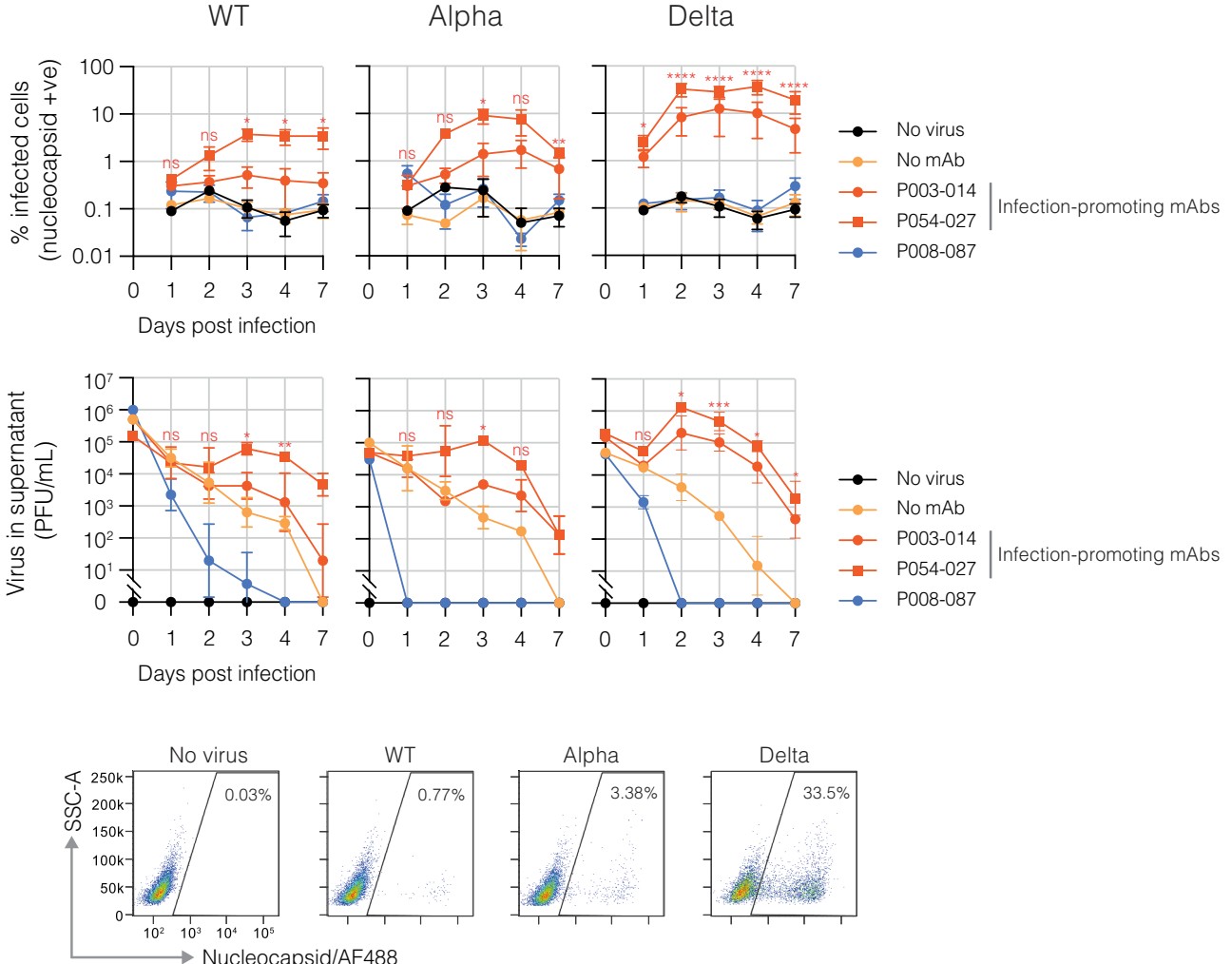

**Fig. 4 | Longitudinal studies of mAb-mediated SARS-CoV-2 infection in monocytes.** Wildtype (WT;B.1), alpha (B.1.1.7) or delta (B.1.617.2) viruses were pre-incubated with infection-promoting mAbs P003_014, P054_027 or non-infection-promoting control mAb P008_087 at 3 µg/ml prior to addition to THP-1 cells at an MOI of 1. Time points were taken at 1, 2, 3, 4 and 7 days post-infection for the determination of % infected cells by flow cytometry (upper panel) and at the time of infection (day 0) and 1, 2, 3, 4 and 7 days post-infection for the determination of infectious virus in the supernatant by plaque assay (lower panel). Results are presented as means from three independent experiments, error bars ±SD. Two-way ANOVA with Tukey's multiple comparison was used to determine whether infection was significant in the presence of P054_027 mAb vs no mAb for each virus and each time point. >0.1 (ns), <0.1 (*), <0.01 (**), <0.001 (***), <0.0001 (****). Representative flow cytometry plots are shown for cells infected with no virus (control), WT (B.1), alpha (B.1.1.7) and delta (B.1.617.2) virus in the presence of P054_027 mAb on day 3 post-infection.

## Infection of iPSC- and primary monocyte-derived macrophages by SARS-CoV-2

Due to their phenotypic similarity to tissue-resident macrophages[50–53] and scalable supply, initial experiments were performed in human induced pluripotent stem cell (iPSC)-derived macrophages. In the presence of mAb P008_015, we observed a modest but reproducible and dose-dependent infection (Fig. 5a)—paralleling results with pseudovirus and THP-1 cells shown in Fig. 2—in which infection peaks at the mAb concentration of 0.55 µg/ml (1.67 µg/ml for delta pseudovirus on THP-1 cells). No infection was seen in the absence of mAb. The high density of cells surrounding foci of infection suggested a chemotactic response by bystander cells (Supplementary Fig. 2b with additional images shown in Supplementary Fig. 2c).

Using similar experimental conditions, these results were extended to primary human monocyte-derived macrophages. Five infection-promoting mAbs (P003_014, P054_027, P008_015, VA14_001 and P054_003; see Figs. 1a, b and 3a) were tested alongside two non-infection-promoting mAbs (P008_087 and P008_056; see Figs. 1a, b and 3a). Given the chemotactic element suggested an inflammatory response driven by infection of iPSC-derived macrophages, experiments were also performed in the presence of the JAK1/2 inhibitor ruxolitinib. As seen previously in both undifferentiated and differentiated THP-1 cells, no infection was seen for virus in the absence of mAb, in the presence of non-infection-promoting mAbs P008_087 and P008_056, or for mAb-only controls (Fig. 5b). Several of the infection-promoting mAbs resulted in significant levels of infection in the absence of ruxolitinib, with P008_015 mediating a mean infection of 3.06%. In the presence of ruxolitinib, however, infection levels increased up to 40-fold, peaking at 49.3% for mAb P008_015, with all infection-promoting mAbs mediating significant levels of infection (Fig. 5b). The magnitude of infection with the different mAbs tracked with their potency on THP-1 cells (see Fig. 1b).

Similar to the staining pattern in differentiated THP-1 cells (Supplementary Fig. 2a), the virus was detected throughout the cytoplasm but not in the nucleus, consistent with the location of coronavirus replication (Fig. 5c with additional images shown in Supplementary Fig. 3a). Infected cells were detected in foci, indicative of spreading infection. Multinucleate, syncytial cells were frequently identified, as

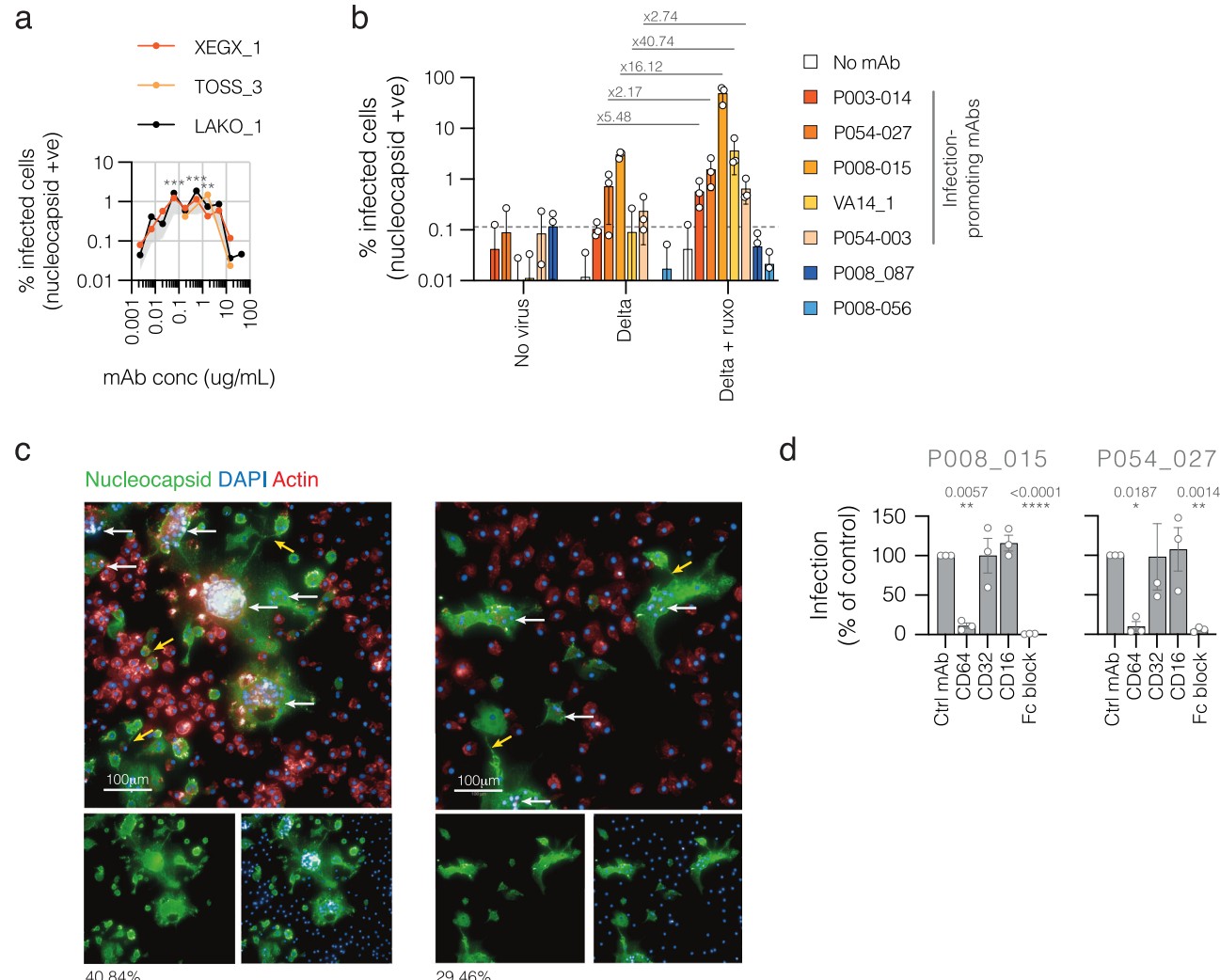

**Fig. 5 | Infection of iPSC- and primary monocyte-derived macrophages by SARS-CoV-2. a** iPSC-derived macrophages were infected with delta virus (B.1.617.2) at an MOI of 1 for 48 hours following pre-incubation with increasing concentrations of infection-promoting mAb P008_015. Cells were intracellularly stained for SARS-CoV-2 nucleocapsid, DAPI and actin, and % infected cells determined by high-content imaging. Results are shown for three iPSC genotypes (XEGX_1, TOSS_3 and LAKO_1). Grey-shaded area represents SEM of all three genotypes. Concentrations of mAb at which the mean % infected cells is significantly greater than virus with no mAb were determined by one-way ANOVA: $p = 0.0002$ at 0.06 μg/ml, 0.0001 at 0.56 μg/ml and 0.0078 at 1.67 μg/ml (>0.1 (ns), <0.1 (*), <0.01 (**), <0.001 (***)). **b** Primary human monocyte-derived macrophages were infected with delta virus at an MOI of 1 following pre-incubation with five infection-promoting mAbs (P003_014, P054_027, P008_015, VA14_001 and P054_003) or two non-infection-promoting mAbs (P008_087 and P008_056) at a concentration of 1 μg/ml, or no mAb as a control, for 48 hours before intracellular staining for SARS-CoV-2 nucleocapsid. All mAbs were also tested in the absence of virus. % infected cells were determined by high-content imaging. Dotted line shows the highest mean value obtained in the absence of virus, thus representing background cut-off. Results are shown as means from three donors, with coloured bars representing the mean and individual data points shown as circles. Error bars ±SD. Fold increase in % infected cells in the presence vs absence of ruxolitinib is shown for each mAb. **c** Detailed images of infected macrophages. Green, SARS-CoV-2 nucleocapsid; red, actin; blue, DAPI. Scale bar 100 μm. Examples of multinucleate structures are indicated by white horizontal arrows; examples of filipodia are indicated by yellow angled arrows. **d** FcγR blocking experiments were performed by pre-incubating macrophages from three different donors with function-blocking mAbs for FcγRI (CD64), FcγRII (CD32) or FcγRIII (CD16), or isotype control mAb, then measuring delta virus infection mediated by mAb P008_015 (left graph) and P054_027 (right graph). % infection was normalised to infection in the presence of control mAb for each donor, then means were derived from normalised infection levels for three different donors, error bars ±SEM. Infection levels in the presence of FcR blocking mAb were compared to control mAb for each condition using two-way ANOVA with mixed effects analyses: >0.1 (ns), <0.1 (*), <0.01 (**), <0.001 (***), <0.0001 (****).

well as filipodia, also staining positive for SARS-CoV-2 nucleocapsid, suggestive of ongoing response to infection by the infected cells themselves (Fig. 5c).

Comparable to results in THP-1 cells (Fig. 1e), CD64 (FcγRI) was predominantly responsible for infection in primary human macrophages (Fig. 5d), with antibody blocking experiments reducing infection levels by over 90% and total Fc block reducing infection to background levels. Blocking CD32 had a variable but non-significant effect on infection levels, while blocking CD16 had minimal effect. Of note, monocytes were isolated by CD14-positive selection, therefore

typically expressed high levels of CD32 and CD64 (Supplementary Fig. 3b) but variable levels of CD16 due to them not being enriched for CD14-low/CD16-high macrophage subsets. Therefore in this context the involvement of CD16 on macrophages that express higher levels cannot be completely ruled out.

To further understand the dynamics of infection in primary human macrophages, the most potent infection-promoting mAb from Fig. 5b, P008_015, was used for full titration experiments using wild-type (B.1), delta (B.1.617.2) and omicron (B.1.1.529.1) viruses and assessed for percent infected cells and production of infectious virus

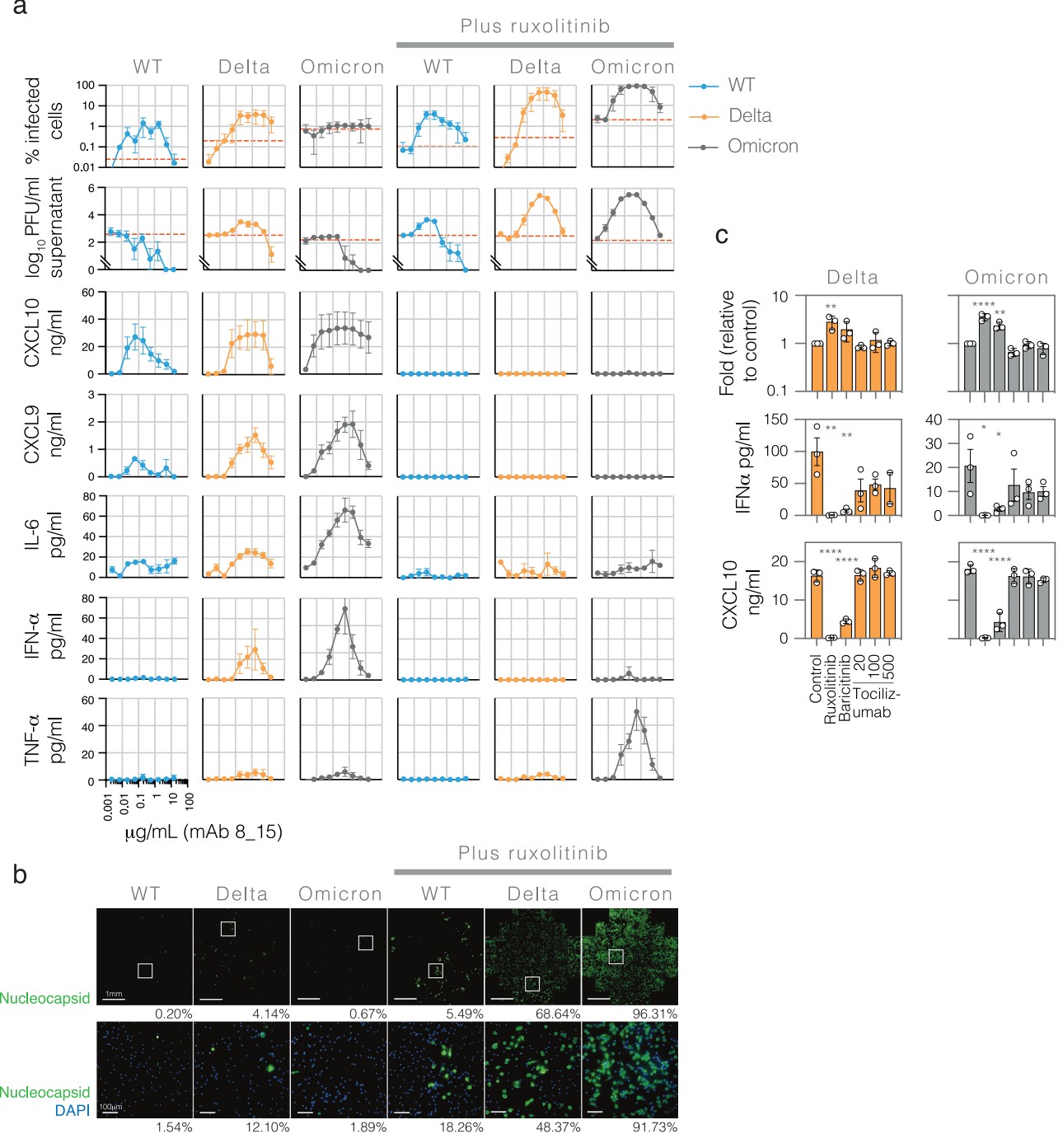

and cytokines in the supernatant (Fig. 6a and b). All three viruses were able to infect the macrophages, with P008_015 having a dose-dependent effect on infection levels. Again, upon the addition of ruxolitinib, infection of macrophages increased significantly, up to 80.7-fold in the case of omicron, leading to mean peak infection levels of 88.8% infected macrophages. Strikingly, the infection levels seen in the presence of ruxolitinib closely mirrored the patterns seen using pseudoviruses on THP-1 cells in the presence of P008_015 mAb (Fig. 2), with infection levels peaking at 0.185 µg/ml for wildtype and 1.67 µg/ml for delta and 0.56 µg/ml for omicron. Similarly, infection levels were lowest for wildtype (peaking at a mean of 4.01%), but higher for delta and omicron (peaking at means of 46.6% and 88.8%, respectively).

Contrary to previous reports, infectious virus was detected in the supernatant of infected macrophages, indicative of productive infection, even in the absence of ruxolitinib. The increase compared to virus detected in the absence of antibody was minor but significant. Importantly, virus in the supernatant decreased after the peak of infection as mAb levels increased, potentially due to active phagocytosis of opsonised virus at the higher, non-infection-promoting concentrations of mAb (Fig. 6a, without ruxolitinib). Infectious virus measured in the supernatant directly correlated with the percentage of infected cells, and this was increased overall following ruxolitinib treatment.

Cytokine measurement by cytometric bead array (CBA) in the supernatant of infected macrophages detected CXCL9 (MIG), CXCL10 (IP10), IL-6, IFNα and TNFα (Fig. 6a), but not IL-10, IL-1β, IL-12 or IFNγ (Supplementary Fig. 3d). In the case of CXCL9, 10, IL-6 and IFNα, the level of cytokine production in cells untreated with ruxolitinib directly

**Fig. 6 | Infection of primary monocyte-derived macrophages by SARS-CoV-2 leads to pro-inflammatory cytokine production. a** Primary human monocyte-derived macrophages were infected with wildtype (WT;B.1), delta (B.1.617.2) or omicron (B.1.1.529.1) virus at an MOI of 1 following pre-incubation with increasing concentrations of infection-promoting mAb P008-015. Virus-only and mAb-only controls were included in all experiments. 48 hours after infection cells were intracellularly stained for SARS-CoV-2 nucleocapsid, DAPI and actin, and % infected cells determined by high content imaging. Supernatant was removed for the determination of infectious virus (PFU/ml) by plaque assay and the production of five pro-inflammatory cytokines (CXCL9, −10, IL-6, IFNα and TNFα) measured by cytometric bead array (CBA). Additional images from these experiments are shown in Supplementary Fig. 3 and cytokine measurements for IFNγ, IL-1β, IL-10 and IL-12 are shown in Supplementary Fig. 4. *Y*-axis scales are the same for all graphs in the same row, and *x*-axis scales are the same for all graphs, marked as grids. Means were derived from independent experiments performed on three different donors, error bars ±SD. Dashed red lines on the % infected cells graphs indicate the mean level of signal detected in the absence of antibody (i.e. background staining for each experiment). Dashed red lines on the PFU/ml graphs indicate the level of virus detected in the absence of antibody (i.e. residual input virus). **b** Images of infected macrophages are shown for the peak infection of each condition shown in (**a**), directly below each graph. Images in the top row show overviews of the whole imaged well (from a 96-well plate), scale bar 1 mm, for SARS-CoV-2 nucleocapsid only, green; images in the bottom row show the highlighted fields of view indicated by white boxes as a representative example, scale bar 100 μm, for SARS-CoV-2 nucleocapsid, green, and DAPI, blue. % infection for each individual well or field of view is shown below each image. **c** Infection of primary human macrophages with delta and omicron viruses was performed as for (**a**) at a single concentration of P008_015 mAb (2 μg/ml for delta infections and 0.5 μg/ml for omicron), in the presence of ruxolitinib (10 μM), baricitinib (1 μM) or 3 different concentrations of tocilizumab (20, 100 and 500 ng/ml). % infection was normalised to control infections (no treatment) for each donor, then means were derived from normalised infection levels for three different donors, error bars ±SD. IFNα and CXCL10 cytokine concentrations in cell culture supernatants were measured as for (**a**). Means were derived from independent experiments performed on three different donors, error bars ±SD. For all experiments, each condition was compared to control using one-way ANOVA: >0.1 (ns), <0.1 (*), <0.01 (**), <0.001 (***), <0.0001 (****).

correlated with the percentage of infected cells in the presence of ruxolitinib (Supplementary Fig. 4a) for all three variants tested (wildtype (England 02), delta and omicron), suggesting (a) that the degree of infection directly impacts cytokine production and (b) that the production of pro-inflammatory cytokines limits infection. It should be noted that the level of pro-inflammatory cytokine produced did not correlate with the concentration of the mAb; rather, the cytokine production declined at higher concentrations of antibody. Furthermore, the peak of infection and hence cytokine production differed for wildtype (0.185 μg/ml) and delta (1.67 μg/ml) and omicron (0.56 μg/ml). Together these observations exclude the possibility that high-level proinflammatory cytokine production is stimulated via conventional antibody-mediated phagocytosis and subsequent intracellular sensing. Thus, there is a strict requirement for macrophages to be infected for high-level pro-inflammatory cytokine production to occur.

We extended the findings with ruxolitinib by using an alternative JAK 1/2 inhibitor, baricitinib, and further investigated cytokine production using the IL-6 receptor antagonist tocilizumab (Fig. 6c). The effect of baricitinib was similar to ruxolitinib, causing a significant increase in antibody-mediated infection of macrophages by both delta and omicron viruses, and a concomitant significant decrease in the production of IFNα and CXCL10. In contrast, although IL-6 triggers JAK/STAT signalling[54], specifically blocking the IL-6 receptor with tocilizumab had no effect on infection levels. CXCL10 production was similarly unaffected, while a variable but non-significant effect on IFNα production could be observed. Together these results support the limitation of spreading infection by paracrine interferon signalling.

Studies in MISTRG6-hACE2 humanised mouse models infected with SARS-CoV-2 have shown that abortive infection in monocytes/macrophages can be rescued by inflammasome or pan-caspase inhibitors, resulting in the release of infectious virus[25]. Under our experimental conditions in primary human monocyte-derived macrophages, we saw no difference in the proportion of infected cells when treating macrophages with either the caspase-1 and −2 inhibitor VX-755 or NLRP3 inflammasome inhibitor MCC950 (Supplementary Fig. 3c). In contrast, treatment with remdesivir, an inhibitor of SARS-CoV-2 replication but not entry, reduced the percentage of nucleocapsid positive cells to background levels (Supplementary Fig. 3d), again confirming de novo production of nucleocapsid rather than phagocytic uptake.

**Polyclonality limits serum-mediated infection of monocytes**

The strict antibody occupancy requirements for antibody-mediated infection to occur in monocytes/macrophages, and its sensitivity to neutralisation, led us to question what would happen in a polyclonal context.

To investigate the possibility of interference between mAbs, we selected a consistently high-level infection-promoting but weakly neutralising mAb from group 1 (P054_027) and combined it with mAbs with a range of binding specificities (Fig. 7a). A minimal effect on P054_027 infection-promoting activity was seen when mAbs from the same binding group were combined (group 1, Fig. 7a). When combined with a weakly neutralising, infection-promoting mAb from group 2 (P054_003), the mAb concentration range over which infection of THP-1 cells occurred was broadened, indicating an additive effect between the mAbs. A potent infection-promoting mAb from group 4 (P008_015) with neutralising activity had a dominant effect over P054_027, with infection closely paralleling the results seen with P008_015 alone. Non-infection-promoting mAbs from groups 4 and 5 (P008_087 and P008_039) with weak or no neutralising activity diminished the infection of THP-1 cells, suggesting that evident mAb-mediated infection of monocytes is context-dependent and easily quashed by dominant antibody activity. The results from these experiments, along with our previous results demonstrating a loss of infection-promoting potential above 50% neutralisation levels, suggest that the existence of dominant infection-promoting activity in polyclonal serum should be rare. To investigate this, a panel of 53 serum samples from 43 individuals infected during the delta wave in the UK (June-July 2021 and December 2021-January 2022; Table 1 and Supplementary Table 1) were tested in THP-1 assays with delta spike-pseudotyped virus, alongside a panel of pre-pandemic samples as negative controls (Fig. 7b). All samples were from acute primary infection, collected between 0 and 22 days post onset of symptoms (POS). As predicted, for the most part, infection of THP-1 cells mediated by sera was low. However, occasional infection was seen with individual serum samples from delta virus-infected individuals, with one sample from an individual with moderate disease promoting high-level infection of THP-1 cells upon dilution. When tested with pseudoviruses bearing D614G spikes, similarly high levels of infection were seen (Supplementary Fig. 4b). Results were summarised as peak fold increase compared to virus only in the absence of sera regardless of the dilution at which this occurred (Fig. 7c). Although an upwards trend was observed in mean peak fold, there was no significant increase with sera from infected individuals compared to pre-pandemic control samples. Normalised areas under the curve (AUC) showed a modest significant increase with sera from infected individuals compared to pre-pandemic control samples (Fig. 7d).

In sum, the presence of neutralising activity in a polyclonal context negates an infection-enhancing effect. However, the ability of diluted sera to induce infection of monocytic cells demonstrates

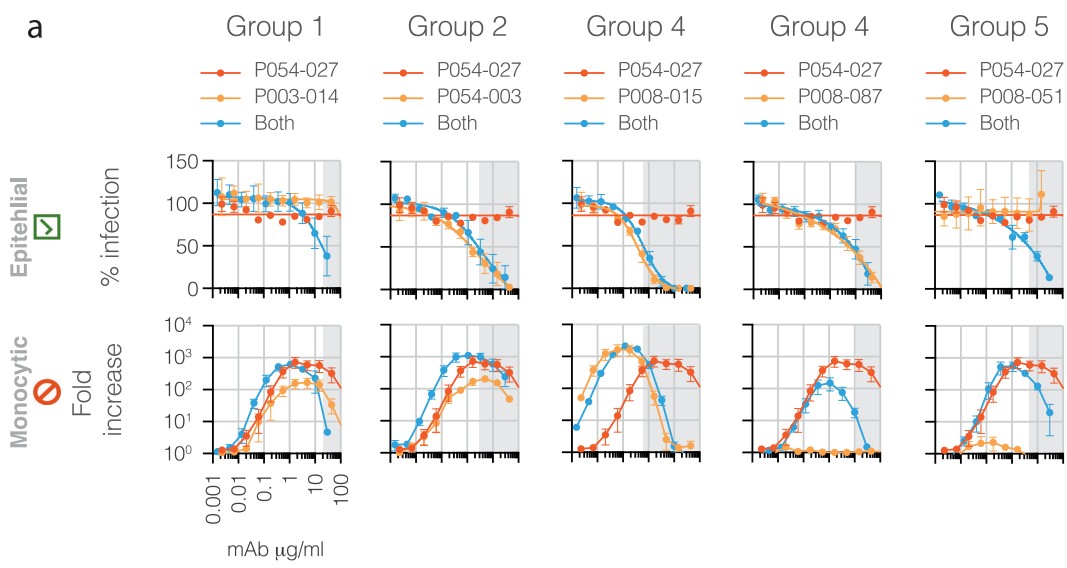

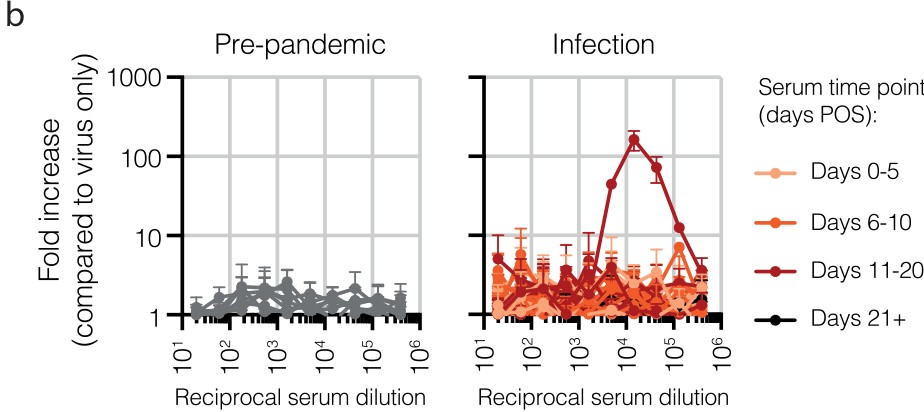

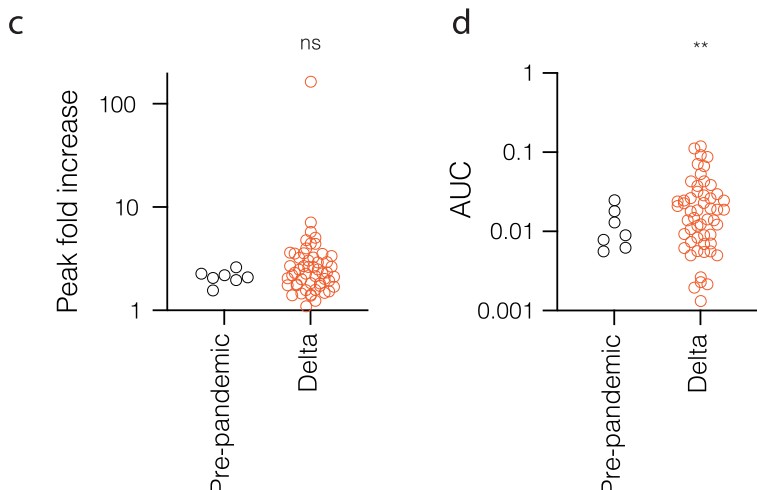

that these antibodies do exist in infected individuals and may impact the course of natural infection and associated pathogenic consequences.

## Discussion

Here we demonstrate efficient productive infection of primary human macrophages, and present a cogent mechanism by which this may occur in primary SARS-CoV-2 infection through antibody-mediated viral entry. Viral replication in macrophages is not limited by abortive replication, but rather innate immune activation in the infected cell that restricts propagation to surrounding cells. Thus we propose that direct macrophage infection by SARS-CoV-2 may contribute to exacerbated pathology of acute COVID-19 when circulating antibodies are suboptimal, or in those with genetic lesions in innate immune sensing

**Fig. 7 | Polyclonality and in vivo relevance. a** mAb mix experiments examining the effect of neutralising, non-neutralising and non-enhancing mAbs on infection-mediating mAbs. Assays were performed in parallel on SARS-CoV-2-permissive epithelial cells (HeLa-ACE2, indicated by green tick) and non-permissive monocytic cells (THP-1, indicated by red 'no entry' symbol). HIV-1-based pseudoviruses bearing SARS-CoV-2 spikes (wave 1 D614G) were preincubated with increasing concentrations of monoclonal antibodies before incubation with cells for 48 hours. Infection is measured by luciferase assay, with % infection and fold increase calculated relative to infection levels in the absence of antibody (in the case of THP-1 cells this is background RLU). Scales on all HeLa-ACE2 and all THP-1 graphs are identical, indicated by grey grid lines. Grey-shaded areas indicate mAb concentrations for mAb mixes at which greater than 50% neutralisation occurs. Overall IgG concentration in the mAb mixes is the same as the individual mAbs, effectively halving the individual mAb concentrations. Means are derived from three independent experiments, error bars ±SEM. **b** Serum samples from uninfected individuals (pre-pandemic negative $n = 7$), and from acute infection during the delta wave in the UK ($n = 53$) were tested in THP-1 assays with delta spike-pseudotypes. Means are derived from two independent experiments, error bars ±SD. Dot plots summarise peak fold increase (**c**) or normalised AUC (**d**) of each serum sample for the different cohorts against delta-pseudotyped viruses. Two-tailed Welch's t-test was used to determine differences between the cohorts. For comparison of peak fold increase, $p = 0.247$ (ns); for comparison of normalised AUC $p = 0.0023$ (**).

## Table 1 | Serum sample details

| DPOS | Severity 0-2 | Severity 3-5 | TOTAL |
|---|---|---|---|
| **0-5** | 8 | 4 | 12 |
| **6-10** | 14 | 6 | 20 |
| **11-20** | 9 | 11 | 20 |
| **21+** | 0 | 1 | 1 |
| **All DPOS** | 31 | 22 | 53 |

53 serum samples from 43 individuals infected during the delta wave in the UK. Samples are categorised by days post onset of symptoms (DPOS) and symptom severity.

and type 1 interferon known to promote disease severity. By contrast, potent polyclonal Ab responses to spike coupled with efficient pattern recognition in the majority of individuals likely act as effective barriers against myeloid cell infection.

This is a clear demonstration of productive SARS-CoV-2 infection of primary human macrophages and a direct association between natural infection of these cells and production of the pro-inflammatory cytokines characteristic of severe COVID-19 (CXCL9 and −10, IFNα, IL-6 and TNFα). Previous models of SARS-CoV-1 and −2 infection of monocytes, macrophages and other FcR-expressing cells have characterised the infection as abortive, with the virus entering cells but unable to complete a replicative cycle[17,25,31,33,34,49]. While abortive infection may occur covertly in infected individuals and contribute to detrimental levels of inflammation, here we have shown that productive infection is possible in primary human macrophages and directly results in the secretion of pro-inflammatory cytokines. The use of ruxolitinib and baricitinib in our primary macrophage experiments, to inhibit JAK1/2-dependent cytokine receptor signalling, revealed efficient and high-level productive infection, with over 88% of cells becoming infected in some experiments. The pan-cytoplasmic nucleocapsid staining, the propensity to detect foci of infected cells, the time of detection (48 hours) and its complete inhibition by remdesivir and FcR blocking are all consistent with FcR-dependent spreading infection. Thus, there is no intrinsic block or cell type-specific deficiency to productive infection in macrophages. Rather, innate sensing of replicating virus triggers the production of pro-inflammatory cytokines, most notably IFNα, and suppresses the spreading infection in bystander cells through paracrine signalling. Ruxolitinib relieves the antiviral state induced by cytokines such as IFNα and promotes secondary infection, as evidenced by up to 80-fold increases in the number of infected cells. This is in line with studies demonstrating the relative susceptibility of early wave variants to inhibition by IFN[55–57]. The direct induction of cytokines downstream of pattern recognition is strongly indicated by the high-level production of CXCL10 in the absence of IFNγ. Furthermore, all cytokines detected directly correlated with the level of infection seen in the macrophages and not with the concentration of mAb, therefore excluding the possibility that they are triggered by conventional antibody-mediated phagocytosis of virus. This is in agreement with reports showing that there is a requirement for direct infection of macrophages engineered

ex vivo to ectopically express ACE2 for substantive cytokine production[28,29], although in this case, we have documented infection of unengineered primary cells. Delays in IFN production[58], intrinsic defects in pattern recognition or IFN signalling[59–62], low expression of IFNAR2[21,63] or the production of anti-IFN autoantibodies[57,64,65], all associated with severe COVID-19, may facilitate the initial infection of monocytes/macrophages in a manner similar to the effects of ruxolitinib, leading to positive feedback loops and detrimental levels of inflammation[11].

Innate sensing of viral infection did not lead to inflammasome activation in our experiments, indicated by the absence of IL-1β detected in the cell culture supernatant and no effect seen following the treatment of cells with caspase and NLRP3 inhibitors. This contrasts with a previous study in human monocytes[17]. However, differences in cell isolation procedures and experimental set-up—including the use of differentiated primary macrophages rather than LPS-primed monocytes—may account for these differences.

It is important to note that we did not see infection of macrophages in the absence of antibody, even in the presence of ruxolitinib, therefore ruling out the possibility that alternative SARS-CoV-2 receptors, or even low levels of ACE2, mediate entry. Nor did we see a positive signal in our microscopy experiments using non-infection-promoting anti-RBD mAbs and infectious virus, thus excluding the possibility that phagocytic uptake of virus is being detected, rather than bona fide infection. The tendency for infected macrophages to form syncytia suggests the occurrence of cell-surface fusion of viral and cellular membranes, rather than viral fusion taking place in endosomes. In this scenario, the antibody-FcR complex substitutes for ACE2, while cell-surface proteases mediate cleavage of the SARS-CoV-2 spike, triggering cell and virus membrane fusion. A similar phenomenon has been modelled for MERS[37]. Additionally, syncytia may be formed by macrophage generation of multinucleated giant cells under inflammatory conditions, thus the circumstances for their formation require further study[66].

Epitope occupancy clearly has a critical bearing on the ability of an individual mAb to mediate SARS-CoV-2 infection, with peak infection for neutralising mAbs occurring most commonly just below the IC50. Logically, this would permit attachment to the target cell while simultaneously leaving some spike molecules free to facilitate fusion. In the case of non- or weakly neutralising antibodies, the concentration at which infection occurs is less constrained and infection occurs at higher antibody doses. This is consistent with the dominance of neutralising activity over promotion of infection and supports the model of antibodies mediating attachment while permitting spike functionality. Of note, antibodies targeting the group 1/class 4 epitope were found to be consistently effective at mediating infection of monocytes/macrophages, which may be partly explained by the necessity for at least two of the three RBD monomers in the trimer to be in the "up" position—the conformation required for engagement with cell surface receptors and subsequent proteasomal cleavage - in order for antibodies to bind to this region[41,42,46,67,68]. Group 2/class 1 antibodies,

which bind to a region overlapping both the group 1/class 4 epitope and the ACE2 binding site, and similarly require the RBD to be in the "up" position, were also effective. In contrast, group 3/class 2 mAbs, which overlap the ACE2 binding site, and NTD mAbs, which bind to a region of spike relatively distal to the target cell, had poor or no ability to mediate infection. Group 4/class 3 mAbs were an interesting group, with some mAbs mediating extremely efficient infection and some none at all. The group 4 epitope is situated on the outer face of the RBD, non-overlapping with the ACE2 binding site, therefore the angle of approach of the individual mAb may be critical here.

Beyond the binding specificity or neutralising capacity of individual antibodies, the importance of competent antibody effector functions in the protection against severe disease has been demonstrated by several studies[6,69–71] and antibody Fc domains play an important role in the balance of adequate and excessive inflammatory responses to infection. For example, afucosylated IgG1 antibodies—which have enhanced binding to FcγRIII and are associated with an inflammatory state—are proportionally higher in individuals with life-threatening COVID-19[72–74] and dengue infection[75,76] compared to those with mild illness. Conversely, class-switching to IgG4—which binds activating FcRs and complement poorly and is associated with a reduced inflammatory state and long-term exposure to antigen—increases following repeated doses of SARS-CoV-2 mRNA vaccine[77–79]. Thus, the role of Fc domain heterogeneity and the relative contributions of FcRs in the promotion or inhibition of macrophage infection requires further investigation. In both our THP-1 and macrophage experiments we implicated FcγRI (CD64) as a key mediator of SARS-CoV-2 antibody-mediated infection. Although our results in THP-1 monocytes, differentiated THP-1 cells, iPSC- and primary monocyte-derived macrophages gave excellent alignment, the possibility that FcγRIII, which has a low affinity for monomeric IgG and is not expressed on THP-1 cells, could play a major role in macrophage infection should not be excluded. Monocytes were isolated in our study by CD14+ selection, therefore enriching for CD16-low and -moderately expressing subsets but excluding those with the highest CD16 expression. In a recent study by Junqueira et al. on primary human monocytes, both FcγRI and -III were shown to mediate infection, with FcγRIIIa (CD16)-expressing monocytes demonstrating the highest susceptibility to infection[17]. In antibody-mediated infection of myeloid cells by dengue virus in mouse models, engagement of FcγRIIIa (CD16) upregulates FcγRIIIa expression, leading to a feedback loop that increases both infection and inflammation[80], thus the possibility of this occurring for SARS-CoV-2 should be explored.

This is a form of antibody-dependent enhancement (ADE), but not in the classical sense[81]. We have described a mechanism by which antibodies of a defined specificity and critical occupancy can expand the tropism of SARS-CoV-2, potentially contributing to exacerbated pathology. This is in contrast to the prototypical example of dengue, in which infection of cells that are already targets of infection is enhanced by antibodies. The in vivo consequences of macrophage infection by SARS-CoV-2 remain to be demonstrated. In terms of clinical outcome, the infection of macrophages may promote clearance of the virus through the production of pro-inflammatory cytokines and recruitment of immune cells to the foci of infection. This may expedite infection resolution. Alternatively, if the recruited cells themselves become infected and diminished in their functional capacity, the outcome may be excessive cytokine production and inefficient resolution of infection, triggering a cycle of increasing disease severity.

The majority of SARS-CoV-2 infections result in mild or asymptomatic disease, with the risk of severe disease decreasing further following vaccination or previous infection. Our results are entirely consistent with this. We have demonstrated the dominant effect of neutralising (and even non-neutralising) antibodies on FcR-mediated infection and identify infection-promoting activity in a minority of serum samples. Therefore, it is not a question of whether we see infection-promoting mAbs or not, but the context in which they exist. In this study, the majority of the work was performed using monoclonal antibodies in order to dissect the mechanisms involved in macrophage infection; however, these do not take into account the complexity of a polyclonal, systemic response to infection. Against a backdrop of neutralising antibodies, for example, in a fully developed polyclonal response to infection, following vaccination or repeat infection, we would hypothesise that infection-promoting antibodies are unlikely to be problematic. Conversely, a suboptimal antibody response produced at the point of peak viral replication in primary infection of an immunologically naïve individual could provide a window of opportunity in which these antibodies may contribute to immunopathology. Although polyclonality would most likely favour viral clearance or other beneficial Fc effector responses[82], conditions for macrophage infection need only be favourable for a short duration to exacerbate a declining clinical status. This is in keeping with the coincidence of detection of humoral responses and deterioration of the clinical condition and highlighted in studies showing a delay in the production of the most potent neutralising antibodies in individuals with severe disease, despite higher overall antibody responses[83,84]. Monocyte/macrophage infection may also be a mechanism for virus dissemination or even persistence through the seeding of long-term reservoirs. We detected filipodia in infected macrophages, indicating the ability of the cells to continue to respond to stimuli while undergoing active parasitisation, which may aid active dissemination of the virus. As macrophages do not represent typical SARS-CoV-2 cellular targets, the dynamics of infection in these cells requires further investigation.

## Methods
### Cells
HEK293T/17 (ATCC CRL 11268™), THP-1 (ATCC TIB-202™) and Vero-E6 (ATCC CRL 1586™) cells were obtained from American Type Culture Collection. Vero-E6 cells were modified to stably express TMPRSS2 by lentiviral vector transduction[85]. The TMPRSS2 expression plasmid was kindly provided by Dr. Caroline Goujon. HeLa-ACE2 cells were kindly provided by James E Voss. HEK293T/17, Vero-E6-TMPRSS2 and HeLa-ACE2 cells were maintained in DMEM supplemented with GlutaMAX and 10% FCS. THP-1 cells were maintained in RPMI 1640 medium supplemented with GlutaMAX and 10% FCS. All cells were grown at 37 °C with 5% $CO_2$.

iPSC lines were kindly provided by the HipSci Consortium and Prof. Fiona Watt. iPSC lines were cultured on vitronectin-coated plates (ThermoFisher, 1 mg/ml) in Essential 8™ Medium (Gibco, UK; #A1517001) supplemented with 1% P/S. Media changes were performed daily and cells were passaged when 70-80% confluent. Production of iPSC-derived macrophages was performed as described[86] with minor modifications. Briefly, cells were washed in PBS and separated into single cells with TrypLE Express (Gibco, UK) for 4 minutes at 37 °C, then washed in Essential 8™ Medium at 220x $g$ for 4 minutes. Cells were resuspended in Essential 8™ Medium supplemented with 50 ng/ml bone morphogenetic protein 4 (BMP-4; Bio-Techne, Minneapolis, USA; #314-BP), 50 ng/ml vascular endothelial growth factor (VEGF; Bio-Techne, Minneapolis, USA; #293-VE), 20 ng/ml stem cell factor (SCF; Bio-Techne, Minneapolis, USA; #11010-SC) and 1 mM rock-inhibitor (Y-27632 dihydrochloride; Enzo Biosciences; #ALX-270-333-M005). Cells were plated at $1.25 \times 10^5$ cells/well in ultra-low adherence 96-well plates (Corning® Costar®, UK) for embryoid body (EB) formation. The plate was centrifuged for 1 minute at 100× $g$ and incubated for 48 hours at 37 °C + 5% $CO_2$ before changing the medium to remove the rock inhibitor but maintain the cytokines in fresh medium, then incubated for a further 48 hours. EBs were then transferred to 10 cm TC-treated dishes coated with 0.1% gelatine and maintained in X-VIVO™ 15 Serum-free haematopoietic Cell Medium (Lonza, UK, #02-060Q) supplemented with 1% GlutaMAX (Gibco, UK), 1% P/S, 0.055 mM β-mercaptoethanol

(Gibco, UK), 100 ng/ml M-CSF (R&D Systems, UK, #216-MC-010/CF) and 25 ng/ml IL-3 (Bio-Techne, Minneapolis, USA; #203-IL). Myeloid precursors were visible in the supernatant after 2–3 weeks. Precursors were harvested every 4–7 days during media changes, using a 40 µm strainer and centrifuging at 250× $g$ for 5 minutes. For final differentiation to macrophages, myeloid precursors were plated in 96-well plates at a density of $1.5 \times 10^4$ cells per well, in the presence of RPMI supplemented with 10% FCS and 50 ng/ml human recombinant M-CSF (R&D Systems, UK, #216-MC-010/CF) for 5 days at 37 °C with 5% $CO_2$ before experimentation. Medium was changed to X-VIVO™ 15 Serum-free haematopoietic Cell Medium (Lonza, UK, #02-060Q) prior to infection.

Peripheral blood mononuclear cells (PBMC) were separated from the peripheral blood of healthy donors (Supplementary Table 2) by density-gradient centrifugation with Histopaque 1077 (Sigma, St. Louis, MO, USA). PBMCs were washed twice with phosphate-buffered saline (PBS) supplemented with 2% FCS. Primary CD14+ monocytes were subsequently isolated by magnetic-activated cell sorting (MACS) with CD14+ Microbeads (Miltenyi Biotech, Bergisch Gladbach, Germany; #130-050-201) according to the manufacturer's protocol, passing cells through fresh LS columns twice. CD14+ monocytes were washed, counted, adjusted to a density of $5 \times 10^5$ cells/ml and plated in 96-well plates, $5 \times 10^4$ cells per well for infection assays or $5 \times 10^5$ cells per well in 12-well plates for cell surface staining, in the presence of RPMI supplemented with 10% FCS and 50 ng/ml human recombinant M-CSF (R&D Systems, UK, #216-MC-010/CF) for differentiation to macrophages for 4–5 days at 37 °C with 5% $CO_2$. The medium was changed to X-VIVO™ 15 Serum-free Haematopoietic Cell Medium (Lonza, UK, #02-060Q) prior to infection.

### Serum samples

Serum samples were collected between the 22$^{nd}$ June and 14$^{th}$ July 2021 and the 16$^{th}$ December 2021 and the 11$^{th}$ January 2022 at St Thomas' Hospital, London, UK[87]. SARS-CoV-2 infection was diagnosed by reverse transcriptase PCR (RT-PCR) of respiratory samples. Disease severity was classified as follows: 0, asymptomatic or no requirement for supplemental oxygen; 1, the requirement for supplemental oxygen (fraction of inspired oxygen ($F_IO_2$) < 0.4) for at least 12 hours; 2, the requirement for supplemental oxygen ($F_IO_2$ ≥0.4) for at least 12 hours; 3, the requirement for non-invasive ventilation/continuous positive airway pressure or proning or supplemental oxygen ($F_IO_2$ > 0.6) for at least 12 hours, and not a candidate for escalation above ward-based (level one) care; 4, the requirement for intubation and mechanical ventilation or supplemental oxygen ($F_IO_2$ > 0.8) and peripheral oxygen saturations <90% (with no history of type 2 respiratory failure) or <85% (with known type 2 respiratory failure) for at least 12 hours; 5, the requirement for extracorporeal membrane oxygenation[87,88]. Delta variant infection was confirmed by whole-genome sequencing of respiratory samples[87].

### Pseudovirus production

Sub-confluent HEK293T/17 cells were transfected in 10 cm dishes with 2 µg of HIV 8.91 Gag-Pol plasmid, 3 µg of CSXW (HIV-firefly luciferase) plasmid and 2 µg of SARS-CoV-2 spike plasmid using 35 µg of PEI-Max (1 mg/ml, Polysciences) and incubated at 37°C[89]. Medium was changed 6–12 hours later and pseudoviruses were harvested 72 hours post-transfection. Following harvest, pseudoviruses were filtered through 0.45 µm filters, DNase treated with 10 U/mL recombinant DNase I (Merck, UK) in the presence of 10 µM $MgCl_2$ for 2 hours at 37 °C. DNase-treated pseudovirus was layered over 20% sucrose in PBS and ultracentrifuged at 116,000× $g$ for 1 hour 30 minutes. Supernatants were aspirated and pellets resuspended in serum-free RPMI-1640 Medium GlutaMAX™ (Gibco, UK).

All spike plasmids used in this study were codon optimised with full-length cytoplasmic tails, with the exception of PMS20 and the wildtype R683G SARS-CoV-2 spike plasmids kindly provided by Paul Bieniasz & Theodora Hatziioannou[45]. SARS-CoV-2 wildtype (B.1) spike was kindly provided by Prof. Nigel Temperton. B.1.1.7 and B.1.351 spikes were described previously[87]. Delta B.1.617.2 and omicron BA.1 spikes were kindly provided by Prof. Wendy Barclay via the MRC G2P Consortium. Spike mutants were generated with Q5® Site-Directed Mutagenesis Kit (E0554) following the manufacturer's instructions[85,90].

### Viruses

The UK SARS-CoV-2 wave 1 reference strain, England 02 (England 02/2020/407073) was obtained from Public Health England. Alpha (B.1.1.7), beta (B.1.351), delta (B.1.617.2), omicron (B.1.1.529.1) were kindly provided by Prof. Wendy Barclay via the G2P Consortium. For propagation, 100 µl of virus was added to confluent Vero-E6-TMPRSS2 cells in 75 cm$^2$ flasks in the presence of DMEM GlutaMAX™ (Gibco, UK) containing 2% FCS. Cells were monitored daily and harvested upon detection of visible CPE. Cultures were filtered through 0.45 µM filters, aliquoted and stored at −80°C before use. Titres were averaged from three independent 6-well plaque assays.

### Plaque assays

For the determination of viral titres, plaque assays were performed in 6-well plates. Virus was 10-fold serially diluted and 500 µl per well added to confluent Vero-E6-TMPRSS2 cells before incubation at 37 °C for 1 hour. 500 µl of pre-warmed overlay (0.1% agarose in DMEM GlutaMAX, supplemented with 2% FCS) was added to each well and incubated for 72 hours at 37 °C. Cells were fixed in 4% formaldehyde for 30 minutes at room temperature, stained with 0.05% crystal violet in ethanol for 5 minutes at room temperature and washed with PBS. Plates were air dried and plaques counted. For the screening of infectious virus in experiment supernatants, plaque assays were performed in 24-well plates, with volumes and quantities scaled down proportionally.

### Neutralisation and ADE assays

Neutralisation assays were performed in HeLa-ACE2 cells[88]. mAbs, serum or plasma (heat inactivated at 56°C for 30 mins prior to first use) were serially diluted in DMEM GlutaMAX™ (Gibco, UK) and incubated with pseudovirus for 1 hour at 37°C. HeLa-ACE2 cells were diluted to 5 ×$10^5$/mL and 50 µl added per well (2.5 ×$10^4$ cells/well) and incubated for 48 hours at 37°C. Parallel THP-1 assays were performed identically, with the exception that all dilutions were performed in RPMI-1640 Medium GlutaMAX™ (Gibco, UK) and cells were diluted to $10^6$/mL (5 ×$10^4$ cells/well). 48 hours after addition of opsonised pseudovirus, cells were harvested for measurement of luciferase signal using Steady-Glo$^R$ Luciferase Assay System (Promega, UK) and a VICTOR™ X Multilabel Reader (Perkin Elmer). Pseudovirus inputs were standardised to give 200,000 RLU/well in HeLa-ACE2 cells at the time of harvest. For assays using serum/plasma this was increased to 400,000 RLU in order to reproducibly capture the more subtle effects of polyclonal sera.

### Infectious virus assays

THP-1 cells were infected at an MOI of 1, using titres pre-determined on Vero-E6-TMPRSS2 cells. Viruses were pre-incubated in serum-free RPMI with 3-6 µg/mL of control non-infection promoting mAb P008_087 or infection-promoting mAbs P003_014 or P054_027, with no-antibody and no-virus controls set up in parallel, for 1 hour at 37 °C in a final volume of 100 µl, prior to addition of cells. Cells were pelleted by centrifugation, resuspended at a density of 2×$10^6$ cells/ml in RPMI supplemented with 10% FCS, and 50 µl (5×$10^4$ cells) added to opsonised virus in 96-well U-bottomed plate. For 72 hour fixed-timepoint assays, mAbs were used at 6 µg/mL. For timecourse assays, in which parallel identical experiments were performed and harvested at 0, 1, 2,

3, 4 and 7 days, mAbs were used at 3 μg/mL. Due to the fact that infection was mediated by antibody, media was not changed after the initial inoculum in order for spreading infection to occur. Upon harvest, supernatant was removed and stored at −80°C for determination of infectious virus production by plaque assay. Cells were fixed in 4% formaldehyde for 30 minutes at room temperature before washing in PBS. Cells were permeabilised in 0.2% Triton X-100 in PBS for 15 minutes at room temperature, then blocked with 5% FCS, 0.1% sodium azide in PBS supplemented with BD Pharmingen™ Human BD Fc block (BD Biosciences #564220, UK) for 20 minutes at room temperature. Blocking solution was removed and cells were incubated with primary antibody - murinised-CR3009[41,91] was used at 2 μg/ml for the detection of intracellular SARS-CoV-2 nucleocapsid - diluted in FACS buffer (PBS containing 1% FCS and 0.1% sodium azide), incubated for 30 minutes at room temperature. Cells were washed twice in FACS buffer then incubated with secondary antibody, donkey anti-mouse IgG (H + L) highly cross-adsorbed secondary antibody, Alexa Fluor™ 488 (Invitrogen, UK, #A21202) diluted 1:500 in FACS buffer for 30 minutes at room temperature followed by an additional two wash steps. Infected cell number was determined by flow cytometry using the BD FACS-Canto II System (BD Biosciences) and FlowJo v10.8.1 (BD) software.

### Differentiated THP-1 assays

THP-1 cells were plated in 96-well plates at a density of $10^5$ cells per well in the presence of RPMI 1640 GlutaMAX™ (Gibco, UK) supplemented with 10% FCS and 100 ng/ml PMA (Promega, UK; #V1171) and incubated at 37 °C for 48 hours. Medium was changed for RPMI 1640 GlutaMAX™ (Gibco, UK) supplemented with 10% FCS and incubated for a further 24 hours. Cells were infected, fixed and stained as per the **Infectious Virus Assay** section above, with the exception that the secondary antibody was goat anti-mouse IgG (Fc-specific)-peroxidase antibody (Sigma #A2554; 2 μg/mL), incubated for 30 minutes at room temperature before washing twice and adding 50 μl per well of TrueBlue peroxidase substrate (SeraCare, UK) substrate for the visualisation of infected cells. Cells were incubated with substrate at room temperature and monitored every 5 minutes for the development of a strong blue stain, at which time substrate was removed, plates air-dried and infected cell number determined on an EliSpot reader using AID EliSpot 8.0 software.

### iPSC-derived and monocyte-derived macrophage infection

Infections of iPSC- and monocyte-derived macrophages (prepared as detailed in **Cells** section above) were carried out in 96-well plates, with all viral and antibody dilutions carried out in X-VIVO™ 15 Serum-free haematopoietic Cell Medium (Lonza, UK, #02-060Q). Viruses were diluted to an MOI of 1 in 50 μl, with mAb or serum diluted as appropriate and 50 μl added to virus. Virus/antibody mixes were incubated for 1 hour at 37 °C. Medium was removed from the macrophages and replaced with the virus/antibody mixes and incubated at 37 °C for 48 hours. Supernatants were removed and stored at −80 °C for future determination of infectious virus titres by plaque assay and production of cytokines by cytometric bead array. Cells were fixed and immunostained according to the **High content imaging** section below.

### Drug treatment and Fc receptor blocking experiments

Inhibitors and drugs were added to cells at the time of infection. Remdesivir (Cayman Chemical Company, US; #30354) was used at a final concentration of 10 μM. The NLRP3 inflammasome inhibitor, MCC950 (Cambridge Bioscience, UK; #17510) was used at a final concentration of 10 μg/ml. The caspase 1 and 4 inhibitor VX-765/Belnacasan (Universal Biologicals, UK; #S2228) was used at a final concentration of 20 μM. JAK1/2 inhibitors ruxolitinib (Cambridge Bioscience, UK; #SM87-10) and baricitinib phosphate (MedChemExpress, UK; #HY-15315A) were used at a final concentration of 10 μM and 1 μM respectively. The IL-6 receptor

antagonist tocilizumab (MedChemExpress,UK; #HY-P9917) was used at final concentrations of 20, 100 and 500 ng/ml. Fc receptor blocking was carried out at room temperature for 1 hour prior to infection. mAbs used for blocking, based on manufacturer's recommendation, were anti-human CD64 clone 10.1 mouse, BioLegend #305008, anti-human CD32 clone IV.3 mouse, Caprico Biotechnology #102724 and anti-human CD16 clone 3G8 mouse, BioLegend #302008. Initial blocking was carried out at 3 μg/ml, then mAbs were replenished upon infection at 0.3 μg/ml. Infection experiments proceeded as detailed in **iPSC-derived and monocyte-derived macrophage infection** and **High content imaging** sections, with the exception that a directly conjugated antibody, Alexa Fluor® 488 anti-SARS-CoV-2 nucleocapsid protein antibody AbCam #ab283243 (used at 1:1000), was used instead of two-step staining for determination of % infected cells.

### High content imaging

Infected iPSC- and monocyte-derived macrophages in 96-well plates (Corning Costar #3596) were fixed in 4% formaldehyde for 30 minutes then washed in PBS. Cells were permeabilised with 0.2% Triton-X100 in PBS for 15 minutes at room temperature, before blocking with 50 μl/well of 5% FCS in PBS supplemented with BD Pharmingen™ Human BD Fc block (BD Biosciences #564220, UK) for 15 minutes at room temperature. 50 μl of 2x murinised-CR3009[41,91] in 0.1% sodium azide was added to the blocking solution to give a final concentration of 2 μg/ml and incubated for 45 minutes at room temperature. Cells were washed twice with wash buffer (PBS containing 0.1% sodium azide and 1% FCS) and incubated with donkey anti-mouse IgG (H + L) highly cross-adsorbed secondary antibody, Alexa Fluor™ 488 (Invitrogen, UK, #A21202) diluted 1:500 in wash buffer for 45 minutes at room temperature. Following two further washing steps, cells were additionally stained for 15 minutes at room temperature with DAPI and CellMask™ Deep Red Actin 647 nm Tracking Stain (Invitrogen #A57245) to highlight nuclei and polymerised/filamentous actin (F-actin) respectively. Following two additional washing steps, cells were stored at 4 °C in PBS containing 1% FCS and 0.5 mM EDTA prior to imaging. The imaging of plates occurred within 3 days of staining.

Images were acquired using the Operetta CLS High Content Analysis System (PerkinElmer) with 20× NA 1.0 water objective. Harmony 4.9 software was used to acquire and process images by creating a quantification analysis pipeline (explained in more detail in Supplementary Fig. 5). In brief, individual cells were identified based on (a) DAPI-stained nuclei and (b) actin cytoplasmic mask that was used to define the cytoplasmic region of each cell. Cells expressing the SARS-CoV-2 nucleocapsid protein (AF488/green signal) within the defined area marked by the cell mask (647 nm/red signal) were considered positive for infection when the threshold for AF488 intensity (SARS-CoV-2 nucleocapsid) was higher than the background intensity levels. This was determined by negative controls (virus with no mAb, mAb with no virus, and no virus with no mAb) in all experiments. Percentages of positive and negative cells were calculated using the quantification analysis pipeline.

### Cell surface receptor staining and flow cytometry analysis

Cell surface Fc receptor expression was determined on THP-1 cells and human monocyte-derived macrophages using the following antibodies: FITC anti-human CD64 antibody, clone 10.1, BioLegend #305006, FITC anti-human CD32 antibody, clone FUN-2, BioLegend #303204, FITC anti-human CD16 antibody, clone 3G8, BioLegend #302006 (5ul per $10^6$ cells as per manufacturer's recommendation). Blocking was performed with BD Pharmingen™ Human BD Fc block (BD Biosciences #564220, UK; 5ul per $10^6$ cells) for 15 minutes on ice before incubation with relevant antibody for a further 30 minutes on ice. Cells were washed

twice in FACS buffer then analysed on a BD FACSCanto II System (BD Biosciences) using FlowJo v10.8.1 (BD) software.

## Cytometric bead array and flow cytometry analysis

Supernatants from primary macrophage experiments were analysed for cytokine content by cytometric bead array (CBA Flex Set, BD Biosciences). Arrays were performed as per the manufacturer's instructions. Briefly, samples were diluted and incubated with a multiplex panel of beads conjugated to antibodies for 9 different analytes (see Supplementary Table 3) for 3 hours at room temperature with regular agitation. Beads were washed twice then incubated with phycoerythrin (PE)-conjugated detection antibodies for a further 2 hours at room temperature then washed twice and analysed using the BD FACSCanto II System (BD Biosciences) and FlowJo v10.8.1 (BD) software. Standards ranging from 0 to 5000 pg/mL were used to determine analyte concentration in the sample from the mean fluorescence intensity (MFI) of each bead cluster, using a four-parameter logistic regression curve. Supernatants were assayed from experiments performed on three different macrophage donors, with results presented as means ± SD.

## Statistics

Statistical analyses were performed using GraphPad FACS v9/10. Unless otherwise stated, all graphs show means from at least 3 independent experiments with errors bars indicating ± SD. For all primary human macrophage experiments, means were derived from at least 3 different donors. Two-way ANOVA was used to compare multiple conditions, with or without Tukey's multiple comparison correction as required. For the comparison of two conditions, t-tests with Welch's correction were used. Normalised areas under the curve (AUCs) were calculated above a threshold of 1 from data normalised to background.

## Ethics

Ethical approval to draw blood from healthy donors as a source for primary CD14+ monocytes was granted by the King's College London Infectious Disease BioBank Local Research Ethics Committee – approvals SN1-100818 and SN1-160322. All volunteers gave written informed consent prior to participation. For the SARS-CoV-2 acute patient samples and pre-pandemic negative samples, surplus serum samples taken as part of routine clinical care were retrieved at the point of being discarded, matched to limited clinical information then anonymised. The ethical oversight for this study was the same as the original studies[87,88], approved by South Central Hampshire B REC (20/SC/0310), with collection of surplus or discarded samples and linked-anonymised clinical information bypassing the requirement for written consent.

## Reporting summary

Further information on research design is available in the Nature Portfolio Reporting Summary linked to this article.

# Data availability

Datasets generated and/or analysed during the current study are included in the paper or are appended as supplementary data. Source data are provided with this paper.

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

## Acknowledgements

This work was funded by: Huo Family Foundation Award to MHM, KJD and SJDN, a Wellcome Trust Senior Research Fellowship to SJDN (WT098049AIA), the MRC Genotype-to-Phenotype UK National Virology Consortium (MR/W005611/1 and MR/Y004205/1to MHM, KJD and SJDN), and MRC Programme Grant (MR/S023747/1 to MHM). EB and MP were supported by the MRC/DBT CARDINNATE consortium grant (MR/V040162/1). NA was funded by the Wellcome Trust PhD programme in Cell Therapies and Regenerative Medicine (108874/Z/15/Z). LBS was funded by MRC grant MR/W025140/1. This work, including the high throughput screening facility "Stem Cell Hotel", was supported by the Department of Health via a National Institute for Health Research comprehensive Biomedical Research Centre award to Guy's and St. Thomas' NHS Foundation Trust in partnership with King's College London and King's College Hospital NHS Foundation Trust. The views expressed are those of the author(s) and not necessarily those of the NHS, the NIHR or the Department of Health and Social Care. We are very grateful to Clara Vila for sharing her time and expertise. We thank members of the Neil lab, past and present, for helpful discussion. We are grateful to members of the G2P Consortium for their generous sharing of expertise and reagents. In particular to Wendy Barclay and team, Greg Towers and Claire Jolly. We thank Theodora Hatziioannou and Paul Bieniasz for the kind provision of the PMS20 and R683G SARS-CoV-2 spike plasmids. We thank George Chennell and the Wohl Cellular Imaging Centre at King's College London for help with high content imaging.

## Author contributions

S.P. and S.J.D.N. designed the study. S.P., H. Winstone and R.P.G. grew the viruses. S.P., H.Wilson, E.B. and M.R.P. performed the neutralisation, monocyte, monocyte-derived macrophage, CBA experiments and high-content imaging. J.S. and C.G. isolated and characterised the monoclonal antibodies. N.A. provided and characterised the iPSC-derived monocytes. L.F. and T.W. set up high-content imaging conditions, L.F. designed high-content imaging analysis pipelines and provided advice. A.M. performed preliminary experiments. H.Winstone constructed spike plasmids and performed mutagenesis. T.A.D.N. provided help with high-content imaging set-up. L.B.S. performed virus sequencing and curated the hospital serum samples. R.P.G., K.J.D., M.H.M. and S.J.D.N. provided funding, advice, discussion and guidance. S.P. wrote the manuscript. All authors edited the manuscript and provided comments.

## Competing interests

The authors declare no competing interests.
