## [Transparent Peer Review file · Nature Communications]

Antibodies to the RBD of SARS-CoV-2 spike mediate productive infection of primary human macrophages

Corresponding Author: Dr Suzanne Pickering

Version 0:

Reviewer comments:

Reviewer #1

(Remarks to the Author)

The manuscript by Pickering et al demonstrates that some monoclonal anti-RBD antibodies facilitate productive infection of human macrophages, which can expand the tropism of SARS-CoV-2 in vitro. Specific Ab groups/classes can facilitate higher or more consistent infection. Macrophage infection is required for inflammatory cytokine production, while blocking JAK1/2-mediated signaling, only in the presence of Ab-mediated entry, increases MP spreading infection, likely through inhibition of cytokine (IFN) production. Lastly, only a small number of human acute polyclonal responses (produced during delta SARS-CoV-2 infections) increase macrophage infection within a narrow window of time post-infection.

Overall, this is a very interesting paper. It uses rigorous well-controlled and orthogonal experimental approaches to show that macrophages can be productively infected in vitro (in contrast to previous reports). The manuscript is very well written. I expect the manuscript will be widely read and impactful due to the fact that previous reports have shown that myeloid cells can be infected in humans and mouse models and can contribute to inflammation, a major driver of SARS-CoV-2 pathology.

Given the reported results, I have a few questions for the authors.

1. First, how should we interpret the importance of this phenomenon, Ab-mediated infection of macrophages, in the current COVID-19 epidemic? Seroprevalence studies in the US and throughout the world suggest that upwards of 95% of individuals (at least in the US) have been infected, vaccinated or both for SARS-CoV-2. Is humoral immunity derived from vaccination or infection sufficient to produce lasting polyclonal Ab responses that would prevent Ab-mediated macrophage infection? The data presented suggests that Ab-mediated entry only occurs in a small number of individuals within a critical window of time during their first infection. Are individuals at risk for Ab-mediated macrophage infection during repeat infections or infection post-vaccination?

2. This manuscript shows for the first time that human macrophages can be PRODUCTIVELY infected. Is productive infection (the release of infectious virions) absolutely required for the generation of inflammation and the release of cytokines? The infection data in the presence and absence of Ruxolitinib suggest that even in the absence of spreading infection that inflammatory cytokines are released. Couldn't abortive infection that involves viral RNA replication and production of viral proteins (different from UV-inactivated or Remdesivir treated which would limit viral RNA replication) without the release of infectious virus be sufficient to drive inflammation?

3. The authors present data showing that Ruxolitinib (JAK1/2 selective inhibitor) can increase spreading infection of macrophages while limiting cytokine production. In patients hospitalized with COVID-19 and receiving dexamethasone who have rapidly increasing oxygen needs and systemic inflammation, the NIH treatment guidelines recommend adding Baricitinib (JAK inhibitor) or Tocilizumab (IL-6 receptor antagonist). Would Baricitinib, similar to Ruxolitinib, increase macrophage infection (by blocking IFN signaling), while Tocilizumab would not?

4. The authors show that Ab-mediated entry into THP-1 is CD64 (FcγRI) dependent (Fig 1E). Is the same true for human monocyte derived macrophages?

Minor comments:

1. How does the cell analysis pipeline handle large, bright syncytia? Including % infection numbers in Supplementary figure

2 images would be helpful. The answer does not diminish the interpretation that infection has occurred, but if the syncytia formation is unequal among the variants, measurement of their % infections may be skewed by their different syncytia formation.

2. Is there an explanation for the TNF α production only in the Omicron + ruxolitinib condition? (Fig 3e)

3. The authors state that the observed syncytia are likely mediated by spike-antibody-FcR interactions. Could macrophage formation of multinucleated cells also contribute?

4. Line 182: Typo, B.1.531 should be B.1.351.

5. Supplementary figure 1D middle graph: I did not notice that this antibody is named anywhere else in manuscript. Is this a new Ab or should it be VA14_041?

6. The Abs listed in Supplementary Figure 3 B-C do not match the figure (mAb 8_15). (line 1188)

7. Supplementary figure 3e does not have a description in the figure legend.

Reviewer #2

(Remarks to the Author)

Pickering et al have put together a manuscript suggesting how SARS-CoV-2 could utilize non-neutralising or sub-optimally neutralising mAbs to infect monocytic cells. This work builds upon previous observations of monocytic infections by SARS-CoV-2 and explores the influence of different classes of Spike targeting mAbs. Overall, the manuscript is well-written and attempts to address the issue using different approaches. The manuscript will benefit from added clarity in the following areas:

Figure 1a: The authors should show the curves for Group 1 mAb VA14_R39 as that mAb is highlighted in the text. It is also unclear why the THP-1 infection curve for Group 7 (SD1) is truncated.

Figure 1e: The surface expression of Fc γ R α s on THP-1 cells vary with age/passage and activation status of the cultured cells, and may impact the efficiency of FcR blocking mAbs. The authors should include a figure describing the surface expression of Fc γ R α s on their THP-1 cells, to aid the reader in understanding the relevance of Figure 1e (particularly the expression of CD32 vs CD64). Details of the FcR-blocking mAbs used (clone, concentration) should also be included in the methods. The authors should also validate or reference (as per manufacturer's recommendation?) that the concentration of FcR-blocking mAbs used (particularly for CD32) is sufficient to abrogate Fc binding.

Figure 2D: The authors propose that the drop in virus particles in the absence of mAbs or presence of P008-087 is driven by clearance of ingested virus by monocytes. Conversely the increase in infectious virus in the supernatant with P003-014 and P054-027 was due to active infection (and an inability to clear ingested virus). This linkage can be better substantiated if the authors can demonstrate other monocytic changes (ROS, pH change), that would be associated with the presence/absence of active viral clearance.

Figure 4B, 4D: The authors should consider breaking up the analysis between early and late infection, to evaluate if THP-1 infections (fold increase / AUC) decline following the increase/maturation of polyclonal antibody responses. The authors should also describe if these infected individuals did seroconvert and develop neutralising responses to Delta, following their Delta infection, and if this rise in neutralising responses inversely correlated with their potential for THP-1 infection by Delta virus.

Discussion: The authors discuss in lines 440-451 how the balance of Fc effector responses may contribute to viral clearance and disease severity. It is also suggested in lines 364-369 that a polyclonal response with good target recognition would likely prevent myeloid cell infection. The authors should discuss or clarify how antibody mediated Fc effector responses work best when activated by a polyclonal pool of antibodies (as compared to individual mAbs) as the antibody polyclonality allows closer Fc-engagements which overall stabilise stronger immune synapses (for better Fc effector responses). This would give the reader context on the limitations of these findings utilizing individual mAbs (lacking polyclonality), and better aligns with the real-world observations that polyclonal antibodies remain protective, and explaining why they are unlikely to cause ADE in the general population.

Minor comments: Authors should check that axis of Figures/Supp Figures are sufficiently labelled for interpretation and details of the tests used are mentioned in the Figure legend (eg: Figure 1C, 1D axis units are in Log? Which test was used? Spearman? Pearson?)

Version 1:

Reviewer comments:

Reviewer #1

(Remarks to the Author)

I am satisfied with the authors' responses to my comments.

Reviewer #2

(Remarks to the Author)

The authors have sufficiently addressed the points raised. The manuscript is ready to be published. This reviewer looks forward to their upcoming body of work promising a deep-dive analysis of wave 1 responses.

Response to reviewers' comments NCOMMS-23-53926-T

Reviewer #1

Reviewer #1 (Remarks to the Author):

The manuscript by Pickering et al demonstrates that some monoclonal anti-RBD antibodies facilitate productive infection of human macrophages, which can expand the tropism of SARS-CoV-2 in vitro. Specific Ab groups/classes can facilitate higher or more consistent infection. Macrophage infection is required for inflammatory cytokine production, while blocking JAK1/2-mediated signaling, only in the presence of Ab-mediated entry, increases MP spreading infection, likely through inhibition of cytokine (IFN) production. Lastly, only a small number of human acute polyclonal responses (produced during delta SARS-CoV-2 infections) increase macrophage infection within a narrow window of time post-infection.

Overall, this is a very interesting paper. It uses rigorous well-controlled and orthogonal experimental approaches to show that macrophages can be productively infected in vitro (in contrast to previous reports). The manuscript is very well written. I expect the manuscript will be widely read and impactful due to the fact that previous reports have shown that myeloid cells can be infected in humans and mouse models and can contribute to inflammation, a major driver of SARS-CoV-2 pathology.

Given the reported results, I have a few questions for the authors.

1. First, how should we interpret the importance of this phenomenon, Ab-mediated infection of macrophages, in the current COVID-19 epidemic? Seroprevalence studies in the US and throughout the world suggest that upwards of 95% of individuals (at least in the US) have been infected, vaccinated or both for SARS-CoV-2. Is humoral immunity derived from vaccination or infection sufficient to produce lasting polyclonal Ab responses that would prevent Ab-mediated macrophage infection? The data presented suggests that Ab-mediated entry only occurs in a small number of individuals within a critical window of time during their first infection. Are individuals at risk for Ab-mediated macrophage infection during repeat infections or infection post-vaccination?

Author response:

Given what we know about the rapidly shifting dynamics of an early antibody response and the likelihood of the initial response in an immunologically naïve individual being mainly weakly or non-neutralising, we hypothesise that antibody-mediated infection of macrophages would be most likely in the first exposure to SARS-CoV-2. Once a neutralising response develops, it will endure or will be rapidly re-deployed by B cell memory in conjunction with T cell responses, therefore we would speculate that the opportunity for antibody-mediated macrophage infection to occur following repeat infection or vaccination would be negligible. We have expanded the last paragraph of the Discussion in order to make this point more robustly.

2. This manuscript shows for the first time that human macrophages can be PRODUCTIVELY infected. Is productive infection (the release of infectious virions) absolutely required for the generation of inflammation and the release of cytokines? The infection data in the presence and absence of Ruxolitinib suggest that even in the absence of spreading infection that inflammatory cytokines are released. Couldn't abortive infection that involves viral RNA replication and production of viral proteins (different from UV-inactivated or Remdesivir treated which would limit viral RNA replication) without the release of infectious virus be sufficient to drive inflammation?

Author response:

The reviewer raises an important point that we are very interested investigating. However, we have tailored our experimental systems towards demonstrating productive infection, and addressing the issue of abortive infection by SARS-CoV-2 unambiguously using a proxy experimental system comes with considerable challenges. As we show in **Figure 3**, in the absence of ruxolitinib we still see macrophages which stain positive for cytoplasmic SARS-CoV-2 N. Thus, in these cells infection must have proceeded through the formation of replication compartments and the production of -ve sense sgRNA templates, and likely further through multiple rounds. Previous data in permissive epithelial cells indicates that major source of type 1 interferon comes via RIG-I/MDA5-mediated sensing of newly synthesized viral RNAs. Such signalling is too late to inhibit viral replication in the infected cell, but limits further rounds of replication through paracrine effects. The major problem to delineate this experimentally in the primary macrophages is that antiviral drugs such as paxlovid or remdesivir block the formation or activity of the viral RdRp in first place, blocking the appearance N+ cells in absence or presence of ruxolitinib, and thus the ligands for cytoplasmic PRRs. It is difficult to tell whether the inflammatory signal emanates only from fully productive replication or also from cells that have yet to reach new viral particle production. To address the question we will need to make clean selective mutants of SARS-CoV-2 that block late stages of viral production that can grow in trans-complementing cells and assayed in macrophages, which is beyond the scope of the study.

We also need to unpick the difference between true abortive infection at an individual cellular level, and clearance of infection (through antibody and phagocytic mechanisms) combined with prevention of future spreading infection at a population level (i.e. within each infected well), as we think that what appears to be abortive infection may actually be the macrophages themselves actively resolving the infection. We have included a figure, below, illustrating this occurring in our experimental system.

Figure 1: Timecourse of infected macrophages, with and without ruxolitinib

Two individual experiments with delta virus in primary human macrophages, with infection mediated by mAb P008_015 on the right and with P054_027 on the left. Experiments were harvested at 20 and 48 hours post infection and macrophages were stained for intracellular SARS-CoV-2 nucleocapsid to determine % infected cells, shown in the graphs above with images of individual wells shown below. Over time, untreated infections stabilise or begin to resolve, while ruxolitinib-treated infections increase, suggestive of a paracrine effect of ruxolitinib on virus infection of new target cells.

We observe that infection levels in the presence and absence of ruxolitinib are similar at 20 hours, then diverge by 48 hours to give the results shown in the manuscript, in which there is a vastly increased level of infection in the presence of ruxolitinib, but the appearance of abortive infection in its absence. Thus we conclude that what we are actually observing is initial productive infection stimulating cytokine secretion, which, through paracrine effects, limits spreading infection to bystander cells, combined with self-limiting infection levels due to the macrophages, in the presence of SARS-CoV-2-specific antibody, responding to the infection within the culture. There is also the added complication that, because antibody is present in the cultures in order for initial and spreading infection to occur, this may also promote phagocytosis of released virions in the same culture, giving the impression of decreased quantities of detectable infectious virus. Thus, depending on the way in which infection is measured, both could be interpreted as abortive infection, and we suspect that this may at least partially explain differences between our study and others.

To conclude, we hypothesise that, yes, abortive infection may well have the potential to drive inflammation. Interestingly, its covert nature makes it more difficult to measure, thus its occurrence would be easy to underestimate. Our intention with this manuscript is not to rule out the possibility of abortive infection occurring, but to demonstrate that productive infection is possible in macrophages and that there is no intrinsic block to complete replication, as has been surmised and, importantly, assumed before. We realise that we may have been too strident on this point and have added a sentence to the Discussion to emphasise that our results still allow for the possibility of abortive infection to occur, and that the two scenarios are not mutually exclusive.

3.The authors present data showing that Ruxolitinib (JAK1/2 selective inhibitor) can increase spreading infection of macrophages while limiting cytokine production. In patients hospitalized with COVID-19 and receiving dexamethasone who have rapidly increasing oxygen needs and systemic inflammation, the NIH treatment guidelines recommend adding Baricitinib (JAK inhibitor) or Tocilizumab (IL-6 receptor antagonist). Would Baricitinib, similar to Ruxolitinib, increase macrophage infection (by blocking IFN signaling), while Tocilizumab would not?

Author response:

Indeed, we did find this to be the case and the extended results lend additional strength to our ruxolitinib observations. We tested ruxolitinib, baricitinib and tocilizumab in parallel and found that ruxolitinib and baricitinib gave similar results, leading to significantly increased infection levels, while infection levels in the presence of tocilizumab were similar to untreated wells. Accordingly, levels of cytokines (CXCL10 and IFN α) produced by infected macrophages were significantly lower in the presence of ruxolitinib and baricitinib, but following tocilizumab treatment remained unchanged relative to untreated infected wells. These results now make up **Figure 3g**.

4.The authors show that Ab-mediated entry into THP-1 is CD64 (FcγRI) dependent (Fig 1E). Is the same true for human monocyte derived macrophages?

Author response:

We have performed FcR blocking experiments in primary human monocyte-derived macrophages and are happy to confirm that, yes, antibody-mediated entry is primarily mediated by CD64. Please see our new **Figure 3d** (and also our new **Supplementary Figure 3b** which details major FcR expression on monocyte-derived macrophages from three donors).

Minor comments:

1. How does the cell analysis pipeline handle large, bright syncytia? Including % infection numbers in Supplementary figure 2 images would be helpful. The answer does not diminish the interpretation that infection has occurred, but if the syncytia formation is unequal among the variants, measurement of their % infections may be skewed by their different syncytia formation.

Author response:

Thank you for this suggestion, we have now included % infection numbers for each image in **Supplementary Figure 2**, and also for **Figure 3** and **Supplementary Figure 3**.

When it comes to syncytia we have improved the pipeline over time. The iPSC-derived macrophages (**Figure 3a** and **Supplementary Figure 2b** and **c**) were more difficult than primary monocyte-derived to image, due to what looked like a dramatic chemotactic response and tendency to cluster after viral exposure. This improved with primary monocyte-derived macrophages and by ensuring that cells were plated at an optimal density. For the determination of infected cells, nuclei are first identified by the Harmony software through DAPI signal, applying the most suitable algorithm and pre-set identification parameters, then cell perimeters are delineated by their actin signal. A cutoff for positive infection is then determined for each experiment, based on 488 signal in uninfected wells. What tends to happen with large syncytia is that the software will split them into individual cells based on their nuclei, as long as there is adequate actin signal. In terms of reflecting the effect of infection on cell numbers, we do not think this is an inaccurate representation, as cells that have become part of a giant syncytium are, to all intents and purposes, infected and will die. Please see below for examples of how the pipeline deals with non- and syncytial cells.

Figure 2: non-syncytial cells

Figure 3: syncytial cells

As we always internally control our experiments i.e. by comparing treatments or antibodies within a given viral variant, and always using cells from at least 3 different donors for each experiment, this should mitigate against any differences between variants. We are currently looking to optimise a pipeline that quantifies syncytia (by size, nucleocapsid staining intensity and number of nuclei

contained within) as a complementary read-out to % infection, as we are interested in conditions that affect their formation (e.g. viral variants, macrophage activation status, presence of immunomodulatory compounds such as ruxolitinib or LPS, and macrophage donor variability).

2. Is there an explanation for the TNF α production only in the Omicron + ruxolitinib condition? (Fig 3e)

Author response:

There is still appreciable TNF α production occurring following infection with delta and omicron viruses in the absence of ruxolitinib, and this correlates with infection levels (see **Supplementary Fig 4a**) and with other cytokines, but it is harder to see due to the linear scales used in **Figure 3e**. We assume that the increased TNF α in the omicron + ruxolitinib conditions is due to increased cell damage caused by the very high level of infection in these wells.

3. The authors state that the observed syncytia are likely mediated by spike-antibody-FcR interactions. Could macrophage formation of multinucleated cells also contribute?

Author response:

Yes this is certainly a possibility, and we have added this to the Discussion (page 12 line 447). As mentioned in Point 1 above, we would like to investigate the formation of syncytia in infected macrophages in more detail.

4. Line 182: Typo, B.1.531 should be B.1.351.

Author response:

Thank you for noticing, this has now been changed.

5. Supplementary figure 1D middle graph: I did not notice that this antibody is named anywhere else in manuscript. Is this a new Ab or should it be VA14_041?

Author response:

Yes correct, it should be VA14_041. Apologies for the error.

6. The Abs listed in Supplementary Figure 3 B-C do not match the figure (mAb 8_15). (line 1188)

Author response:

Again, thank you for noticing and apologies for the error. This has now been corrected.

7. Supplementary figure 3e does not have a description in the figure legend.

Author response:

This has now been rectified.

Reviewer #2

Reviewer #2 (Remarks to the Author):

Pickering et al have put together a manuscript suggesting how SARS-CoV-2 could utilize non-neutralising or sub-optimally neutralising mAbs to infect monocytic cells. This work builds upon previous observations of monocytic infections by SARS-CoV-2 and explores the influence of different classes of Spike targeting mAbs. Overall, the manuscript is well-written and attempts to address the issue using different approaches. The manuscript will benefit from added clarity in the following areas:

Figure 1a: The authors should show the curves for Group 1 mAb VA14_R39 as that mAb is highlighted in the text. It is also unclear why the THP-1 infection curve for Group 7 (SD1) is truncated.

Author response:

Thank you for pointing this out. We have now included mAb VA14_R39 in **Figure 1a**. The THP-1 infection curve for Group 7 (SD1) appeared truncated because it dipped just below 1 on the y-axis. This was also the case for some of the other THP-1 graphs. We have rectified this by making a very minor adjustment to all of the THP-1 graphs, now starting the y-axis at 0.9 to include these points.

Figure 1e: The surface expression of FcγRs on THP-1 cells vary with age/passage and activation status of the cultured cells, and may impact the efficiency of FcR blocking mAbs. The authors should include a figure describing the surface expression of FcγRs on their THP-1 cells, to aid the reader in understanding the relevance of Figure 1e (particularly the expression of CD32 vs CD64). Details of the FcR-blocking mAbs used (clone, concentration) should also be included in the methods. The authors should also validate or reference (as per manufacturer's recommendation?) that the concentration of FcR-blocking mAbs used (particularly for CD32) is sufficient to abrogate Fc binding.

Author response:

We have added in a figure showing cell surface expression of FcγRs (CD16, CD32 and CD64) on undifferentiated THP-1s; this is the new **Supplementary Figure 1e**. We are always careful to use THP-1 cells with minimal passage number, and here it is clear that they express a high level of CD32 and CD64 but no CD16. We have also done the same for primary human macrophages from three donors, and this is now **Supplementary Figure 3b**. We have further strengthened our data implicating CD64 by performing FcR blocking experiments on primary human macrophages from three different donors and again show a clear involvement of CD64. However, we do not think this entirely rules out the involvement of CD16, as we isolated monocytes by CD14 positive selection, thereby enriching for CD16-low and -moderate cells, but not for CD14-low/CD16-high non-classical cells. Furthermore, using Fc block as an additional control completely eliminated infection, suggesting that the minor residual infection seen in the presence of the CD64 blocking antibody might be attributable to another Fc receptor. Furthermore, data for anti-CD32 blocking of infection in primary macrophages were variable. Thus, while we have strongly implicated CD64 in this process, we cannot completely rule out CD16 and CD32 playing minor roles under these culture conditions.

Apologies for the oversight with the identity and concentration of FcR-blocking mAbs, this has now been rectified. All were chosen based on claims by the manufacturer that they blocked function. We were further guided in our choice of antibody by a recent study demonstrating the involvement of CD16 in antibody-mediated macrophage infection by Junqueira et al. Note that for CD32 we used two different antibody clones: FUN-2 and IV.3. Both gave similar results so we have only included those from IV.3 in the manuscript. We used the antibodies at a concentration at least 10x over the recommended concentration for cell staining, which is already recommended in excess.

Figure 2D: The authors propose that the drop in virus particles in the absence of mAbs or presence of P008-087 is driven by clearance of ingested virus by monocytes. Conversely the increase in infectious virus in the supernatant with P003-014 and P054-027 was due to active infection (and an inability to clear ingested virus). This linkage can be better substantiated if the authors can demonstrate other monocytic changes (ROS, pH change), that would be associated with the presence/absence of active viral clearance.

Author response:

We have changed the sentence from “likely” to “potentially” [p.6, line 226]. To clarify, we do not propose that the drop in virus particles in the *absence* of mAbs is necessarily due to clearance. This could be due to natural decay of virus in the supernatant over time, as stated. The point we are making is that infection-promoting mAbs have a dramatically different impact on the infection of monocytes compared to non-infection promoting mAbs or no mAb, or even infection-promoting mAbs at different concentrations. We thank the reviewer for the interesting suggestion and will certainly take this point on board for future studies. However, we do not think that the point we’re making requires us to prove active clearance of the ingested virus; rather that we have used exceptionally good controls by including antibodies that, while crucially still binding to the SARS-CoV-2 spike, promote a dramatically different outcome to the infection-promoting antibodies we have carefully characterised. We have shown this time and again throughout the manuscript, always using non-infection promoting mAbs as a control alongside no mAb to ensure that in no way can any of the results we have shown be mis-interpreted as antibody-mediated virus uptake rather than *bona fide* infection. We have done this very successfully in several different systems: THP-1 cells with pseudoviruses, THP-1 cells with live infectious virus, differentiated THP-1 cells with live infectious virus, iPSC-derived macrophages with live infectious virus and last but certainly not least, primary human monocyte-derived macrophages from multiple different donors with live infectious virus. The fact that (i) we detect more infectious virus in the supernatant than we put in, (ii) this occurs after an eclipse phase, (iii) the nucleocapsid staining pattern of infected macrophages matches expected subcellular locations of SARS-CoV-2 replication, (iv) we frequently see the formation of syncytia in infected, but not uninfected, cultures, and crucially (v) that all of these observations are abolished by the addition of a small molecule inhibitor of the viral polymerase (remdesivir), provide a very solid argument for active infection of a cell type that is not a typical target of SARS-CoV-2. We do not think that showing active monocytic changes would further substantiate this point in this instance, especially if we see active monocytic changes in infected cultures, which is likely.

Figure 4B, 4D: The authors should consider breaking up the analysis between early and late infection, to evaluate if THP-1 infections (fold increase / AUC) decline following the increase/maturation of polyclonal antibody responses. The authors should also describe if these infected individuals did seroconvert and develop neutralising responses to Delta, following their Delta infection, and if this rise in neutralising responses inversely correlated with their potential for THP-1 infection by Delta virus.

Author response:

Again, we thank the reviewer for this useful point and can assure them that we have considered this at length. Our purpose with this figure is to demonstrate that antibody-mediated infection is possible in polyclonal sera, but that we detect it at high level only occasionally. We completely take the reviewer’s comments on board, and are currently conducting a much more thorough investigation of longitudinal plasma samples obtained from individuals infected in wave 1 of the pandemic, all of which have been characterised for SARS-CoV-2 antibody binding and neutralisation titres, ADCC, cytokine levels etc. for a more focused study on the propensity for Ab-mediated infection to occur in a polyclonal context.

Discussion: The authors discuss in lines 440-451 how the balance of Fc effector responses may contribute to viral clearance and disease severity. It is also suggested in lines 364-369 that a polyclonal

response with good target recognition would likely prevent myeloid cell infection. The authors should discuss or clarify how antibody mediated Fc effector responses work best when activated by a polyclonal pool of antibodies (as compared to individual mAbs) as the antibody polyclonality allows closer Fc-engagements which overall stabilise stronger immune synapses (for better Fc effector responses). This would give the reader context on the limitations of these findings utilizing individual mAbs (lacking polyclonality), and better aligns with the real-world observations that polyclonal antibodies remain protective, and explaining why they are unlikely to cause ADE in the general population.

Author response:

We appreciate the point the reviewer makes and we have tried to be very clear that a mature polyclonal response engendered by either vaccination or prior infection is unlikely to mediate this effect to a pathogenic level because there is no evidence of a classical ADE response of the kind observed with dengue virus, when individuals are exposed to a different serotype. We feel this is particularly important to state because of misuse of such un-caveated data by those who want to cast vaccines as dangerous.

In addition to the above sections specifically mentioned by the reviewer, we had also stated in the "Limitations" section of our study that the majority of our work had been conducted with monoclonal antibodies, and further work is required to understand this in a polyclonal context. We have now removed the "Limitations" section, therefore this point has been made more robustly in the Discussion. We have also stated more clearly in the last paragraph of the Discussion that, even if infection-enhancing mAbs are detected in individuals, our hypothesis is that antibody-mediated infection would not happen against a backdrop of neutralising antibodies; therefore in most infections, and following vaccination or repeat infection, the chances of this occurring would be minimal.

However, we would like to emphasise that the balance only needs to be in favour of mediating infection for a transient period for macrophages to become infected, and that this may be sufficient to trigger a cycle of increased inflammation and increasing disease severity. The stabilisation of an immune synapse through polyclonal antibody-Fc engagements would not apply in the scenario of antibody-mediated infection of macrophages by virus vs antibody-mediated phagocytosis of virus. In this case, it is not about stabilising an immune synapse but rather the balance of mAbs bound to a virus that allow infection to occur in a single instance, particularly if cell-surface fusion of the virus and macrophage is involved.

Thus, while of course we agree that polyclonal antibodies generally would be protective, and have further emphasised this in our Discussion, we maintain that there may be circumstances that favour infection of macrophages rather than viral clearance, particularly during very early responses to SARS-CoV-2 in naïve individuals. The fact remains that this is exactly the period of highest risk of the development of severe disease – not early infection, when virus replication is rampant and unchecked by an antibody response, but after the antibody response becomes detectable.

Minor comments: Authors should check that axis of Figures/Supp Figures are sufficiently labelled for interpretation and details of the tests used are mentioned in the Figure legend (eg: Figure 1C, 1D axis units are in Log? Which test was used? Spearman? Pearson?)

Author response:

Apologies for any omissions, we have now rectified this.